# GWAS of adventitious root formation in roses identifies a putative phosphoinositide phosphatase (*SAC9*) for marker-assisted selection

David Wamhoff[1], Laurine Patzer[2], Dietmar Frank Schulz[2¤], Thomas Debener[2], Traud Winkelmann[1]*

**1** Institute of Horticultural Production Systems, Section Woody Plant and Propagation Physiology, Leibniz Universität Hannover, Hannover, Germany, **2** Institute of Plant Genetics, Section Molecular Plant Breeding, Leibniz Universität Hannover, Hannover, Germany

¤ Current address: Federal Office of Consumer Protection and Food Safety, Braunschweig, Germany
* traud.winkelmann@zier.uni-hannover.de

**Data Availability Statement:** All data are available at the Research Data Repository of Leibniz University Hannover under the following DOIs (The DOIs are not yet publicly available, but if the

## Abstract

Rose propagation by cuttings is limited by substantial genotypic differences in adventitious root formation. To identify possible genetic factors causing these differences and to develop a marker for marker-assisted selection for high rooting ability, we phenotyped 95 cut and 95 garden rose genotypes in a hydroponic rooting system over 6 weeks. Data on rooting percentage after 3 to 6 weeks, root number, and root fresh mass were highly variable among genotypes and used in association mappings performed on genotypic information from the WagRhSNP 68 K Axiom SNP array for roses. GWAS analyses revealed only one significantly associated SNP for rooting percentage after 3 weeks. Nevertheless, prominent genomic regions/peaks were observed and further analysed for rooting percentage after 6 weeks, root number and root fresh mass. Some of the SNPs in these peak regions were associated with large effects on adventitious root formation traits. Very prominent were ten SNPs, which were all located in a putative phosphoinositide phosphatase *SAC9* on chromosome 2 and showed very high effects on rooting percentage after 6 weeks of more than 40% difference between nulliplex and quadruplex genotypes. *SAC9* was reported to be involved in the regulation of endocytosis and in combination with other members of the *SAC* gene family to regulate the translocation of auxin-efflux *PIN* proteins via the dephosphorylation of phosphoinositides. For one SNP within *SAC9*, a KASP marker was successfully derived and used to select genotypes with a homozygous allele configuration. Phenotyping these homozygous genotypes for adventitious root formation verified the SNP allele dosage effect on rooting. Hence, the presented KASP derived from a SNP located in *SAC9* can be used for marker-assisted selection in breeding programs for high rooting ability in the future.

reviewers would like to study them, we will be happy to provide them with an account): Wamhoff et al. (2023): PhenotypicRawData_GWAS_KASP. xlsx. https://doi.org/10.25835/mto1o61o Wamhoff et al. (2023): KASP_raw_Fluorescences.xlsx. https://doi.org/10.25835/8vfpmh7w Wamhoff et al. (2023): GenotypicData_Tetraploid_SNPChip.csv. https://doi.org/10.25835/ie9cdzgi.

**Funding:** TD: Aif Projekt GmbH (FKZ: KF2554805MD4), https://www.aif-projekt-gmbh. de/ TD: Aif Projekt GmbH (FKZ: ZF4619701MD8), https://www.aif-projekt-gmbh.de/ TD: Bundesministerium für Ernährung und Landwirtschaft (BLE) (FKZ: 2819HS010), https:// www.ble.de/ This study was carried out with financial support from Leibniz University Hannover. The funders had no role in study design, data collection and analysis, decision to publish, or preparation of the manuscript.

**Competing interests:** The authors have declared that no competing interests exist.

## Introduction

The genus *Rosa* L. comprises 30,000 to 35,000 cultivars within approximately 200 species [1, 2]. Despite this diversity, only 7 to 15 wild species have contributed to the current modern rose cultivars [2]. The ploidy of roses ranges from 2x to 8x, while most modern hybrid cultivars are tetraploid [3]. Within different horticultural segments, roses are of great importance, e.g., as garden plants, pot plants, and especially cut flowers [3]. In addition to their usage as ornamentals, roses are also essential in the production of pharmaceutical and cosmetic products, making the genus *Rosa* one of the most important genera within the huge family of Rosaceae [3, 4].

Due to their high heterozygosity and self-incompatibility, commercial rose cultivars are mainly propagated vegetatively [2, 4, 5]. Generative propagation, on the other hand, is used for the propagation of wild species and rootstocks as well as in breeding activities [2, 4, 5]. Depending on the rose type, vegetative propagation takes place in different ways: comparatively easy-to-root miniature pot roses are mainly propagated by cuttings, while cut roses are propagated either by stenting, grafting, or cuttings, and garden roses are mainly propagated by budding and grafting [6, 7]. However, as xenovegetative propagation methods are laborious, costly, and require trained specialised personnel, propagation via more efficient multiplication by cuttings is desirable and of increasing importance to rose breeders and producers [6–8].

Adventitious root (AR) formation after excision is a prerequisite to the success of and a limiting step in propagation via cuttings [9, 10]. However, modern rose cultivars are characterised by strong genotypic differences in the ability for AR formation in general or appropriate quality and number of roots in particular [7, 11]. AR formation is a complex process regulated by hundreds of different genes and affected by numerous exogenous factors within different developmental phases [10, 12, 13]. It can be divided into three to four distinct phases. Starting with the optional phase of dedifferentiation, in case no competent cells are present, the three main phases of AR formation are the induction phase (cellular signalling and programming), the initiation phase (cell division and formation of cell clusters), and the expression phase (root primordium growth and root emergence) [10, 12]. An outstanding role in the regulation of AR formation is attributed to genes associated with auxin biosynthesis, transport and signalling or genes regulated by auxin [14, 15]. Examples for auxin-related genes described in the literature to be involved in AR formation include auxin transport-associated genes of the *AUX*, *LAX*, and *PIN* families [16, 17], auxin responsive genes such as *AUX/IAAs* and *ARF* family genes, or auxin biosynthesis-related *YUCCA1* and *SUR1/2* [18]. Additionally, impacts of genes associated with further plant hormones such as ethylene (e.g., *AP2/ERF*), cytokinins, strigolactones, and jasmonic acid, especially during early phases of AR formation, were described [15, 19–22]. Genes related to carbohydrate processing and transport (e.g., *SUS*, *SUC*) as well as cell division processes (e.g., *LBD* genes, *BBM*) also play pivotal roles in different phases of AR formation [23–26]. Moreover, environmental factors such as the nutrient and carbohydrate status of stock plants, nodal position on the mother plant, and light conditions complement the complex regulation of the process of excision-induced AR formation [10, 27–29]. However, in general, and in the case of the genus *Rosa*, further research is needed to identify the genetic causes of genotypic differences in AR formation and to enable more feasible and economic propagation of important horticultural and forestry species by cuttings.

Research on complex traits such as AR formation in roses is facilitated by the availability of numerous molecular markers and several genomes published since 2017 [30–34]. Specifically, the WagRhSNP 68 K Axiom SNP array for roses covering 68,893 single nucleotide polymorphisms (SNPs) enables genome-wide association studies (GWAS) through high-density marker analysis to identify SNPs and genomic regions associated or linked to the regulation of

the traits of interest [35]. To date, GWAS has been used in rose to study several economically important traits, namely, flower traits such as petal number [30, 36] and size [36], anthocyanin and carotenoid content of petals [37], fragrance [36], prickle density [30], and regeneration traits such as callus formation [38], adventitious shoot regeneration [39], and AR formation [40]. Some of the association mappings were followed by the development of molecular markers, namely, *Kompetitive Allele Specific PCR* (KASP) markers, that allow for the selection of good performing genotypes in breeding programs by distinguishing different allele configurations for the SNP of interest by fluorescence signal quantification [36, 40].

Nguyen et al. [7] studied AR formation for a diversity panel of 95 garden roses and observed high genetic variation, both under *in vitro* and *in vivo* conditions. The authors identified significantly associated SNPs for different AR-related traits located close to gene homologues known to be involved in AR formation, such as *ARF19*, *EIN-LIKE3*, and *WOX8* [7]. In the present study, the work of Nguyen et al. [7] on excision-induced AR formation in roses using a GWAS approach was extended by including a markedly higher number of genotypes (190 compared to 95) and testing a greater number of cuttings per genotype. The second objective of the present study was to use the GWAS outcomes to develop a KASP marker assay for marker-assisted breeding and to test this KASP marker in an independent set of genotypes. The KASP marker could then be used in further breeding programs aiming to improve rooting performance in rose.

## Materials and methods

### Plant material

A set of 95 cut roses (ST1A in S1 Table) and 96 garden rose genotypes (previously described by [37], ST1A in S1 Table) was used for the GWAS on traits related to AR formation. The garden rose genotypes were cultivated as greenhouse-grown stock plants grafted onto *Rosa corymbifera* 'Laxa' at the Federal Plant Variety Office, 30627 Hannover, Germany. Cut rose genotypes were cultivated as own-rooted greenhouse-grown stock plants at Rosen Tantau, 25436 Uetersen, Germany. For KASP marker validation, leaf tissue was collected from 377 genotypes available as field-grown plants grafted onto *Rosa corymbifera* 'Laxa' at the Europa-Rosarium Sangerhausen, 06526 Sangerhausen, Germany (S2 Table).

### AR formation experiments

*In vivo* rooting of two-nodal, semi-hardwood cuttings was conducted in a hydroponic rooting system previously described by [7]. The cuttings were inserted into 63 holes drilled into a non-transparent plastic plate (42*58*0.3 cm) (S1 Fig). Therefore, the lower leaf was removed while the upper leaf remained. The plates were placed onto black plastic trays (41.5*56.5*9 cm). The trays were filled up to 1 cm below the top with tap water, which was replaced by 0.13% fertiliser solution (NPK 15-10-15 + 2 MgO, Planta Düngemittel GmbH, 93128 Regenstauf, Germany) after two weeks of cultivation. The solution in the rooting trays was permanently aerated with fish tank pumps (90 L h$^{-1}$). Rooting experiments were conducted for six weeks in total in a foil tent within a climatised greenhouse chamber (18°C setting temperature) and under a photoperiod of 16 h, which was realised by additional lighting provided by SOD AGRO 400–230 high pressure sodium lamps (DH Licht GmbH, 42489 Wülfrath, Germany) if global irradiance (detected outside the greenhouse) fell below 35 klx. Temperatures inside the foil tent were measured with a LOG 32 TH PDF data logger (Dostmann electronic GmbH, 97877 Wertheim-Reicholzheim, Germany) and are presented in S3 Table.

Cuttings of garden rose genotypes were rooted in two independent experiments from 22/06/2020 to 03/08/2020 (GR1) and from 16/09/2020 to 28/10/2020 (GR2). Likewise, cut roses

were rooted in two independent experiments from 16/03/2021 to 27/04/2021 (CR1) and from 05/07/2021 to 17/08/2021 (CR2). A selection of eleven (CR1) or 13 (CR2) garden rose genotypes was included as a reference within cut rose rooting experiments (ST1A in S1 Table). Within a randomised complete block design, three cuttings each were tested in three blocks per experiment and genotype, resulting in phenotyping of a total of 18 cuttings per genotype. However, due to limitations in the number of available mother plants, the intended 18 cuttings per genotype could not be reached in a few cases (for the exact cutting numbers, please see ST1A in S1 Table).

For KASP marker validation, cuttings from 43 homozygous genotypes were selected and rooted in the same setup as in the GWAS phenotyping experiments. Within two independent experiments from 15/06/2022 to 27/07/2022 (KASP1) and from 14/07/2022 to 25/08/2022 (KASP2), ten cuttings per genotype and experiment were rooted. They were equally distributed over two blocks, resulting in 20 tested cuttings per genotype. The number of tested cuttings per genotype varied between ten and 20 cuttings (S10 Table).

## AR formation phenotyping and data processing

AR formation (yes/no) was evaluated nondestructively after 3, 4, and 5 weeks. After 6 weeks, the root numbers and root fresh masses were collected in addition to determination of AR formation (yes/no) in a destructive final evaluation. Cuttings that started to rot were removed from the hydroponic rooting system to prevent contamination. Weekly, AR formation percentages were calculated for the number of surviving cuttings per block, which were used to calculate weighted means using the R function *weighted.mean* [41] to account for different replicate numbers. Pearson´s correlation coefficients were calculated between AR formation percentages after 3, 4, 5, and 6 weeks of rooting, as well as for root number and total root fresh mass per rooted cutting, and the calculated average root fresh mass per root.

## Marker–trait association analyses

The 68 K WagRhSNP Axiom array [35] was used to analyse SNPs for the garden (previously introduced by [37]) as well as the cut rose populations. Tetraploid allele dosages were called using the R packages SNPpolisher [42] and *fitTetra* [43]. Tetraploid recorded SNP data were used for GWAS analysis performed within the R package *GWASpoly* [44]. In the GWAS analyses, phenotypic data were used as ordered quantile normalised transformed data obtained via the R function *orderNorm* [45]. GWAS analyses were performed under the assumption of a minor allele frequency (MAF) of 0.05 and a marker missing threshold of 0.1, excluding all markers that showed missing values for more than 10% of the analysed genotypes. This resulted in 25,333 SNPs located in 18,051 contigs to be included in the association mappings. The kinship leave-one-chromosome-out (*K.loco*) method [46] and two principal component analyses implemented in *GWASpoly* were used as covariates to consider population structure, which was evaluated based on quantile–quantile (QQ) plots. Applying a significance level of $p<0.05$, using the *M.eff* method in *GWASpoly*, considering linkage disequilibrium (LD) and using the Bonferroni adjustment [47], the significance threshold was defined at $4.5*10^{-6}$ ($-\log_{10}(p) = 5.35$). Association mappings were conducted assuming an additive (add) allele dosage-trait relationship and a simplex-dominance model (1-dom) assuming the reference allele (1-dom-ref) or the alternate allele (1-dom-alt) as dominant [44].

In addition to significantly associated SNP markers, genomic regions of interest showing nonsignificant but distinctive peaks were analysed. Peaks were selected based on the following criteria: low distance to the significance threshold, low distance of the single SNPs within the SNP cluster to each other, and appearance of the peak for several AR formation traits.

Identified peaks were analysed in more detail to detect and select SNPs with large effects on AR formation traits, namely, AR formation percentages after 6 weeks, root number, and root fresh mass per rooted cutting. This was performed in a four-step pipeline: (1) Only SNPs with at least four out of five possible allele dosage groups (ADGs) and at least eight genotypes per ADG were taken into account. (2) Selected SNPs were tested for significant differences dependent on allele configuration via the Kruskal–Wallis test ($p>0.05$). (3) Significant SNPs were analysed by post hoc Fisher´s least significant difference criterion (LSD) to determine which ADGs differed significantly from each other ($p>0.05$, Holm–Bonferroni adjustment). (4) Additive and dominant trait effect sizes were calculated between certain ADGs for selected SNPs that showed significant value differences for the traits of interest as follows:

Additive effect: $E_{add} = |\bar{x}(ADG_{max}) - \bar{x}(ADG_{min})|$

Simplex dominance effect for the reference allele: $E_{1\text{-}dom\text{-}ref} = |\bar{x}(ADG_{max}) - \bar{x}(ADG_k)|$, where $k$ is $\forall\, x \in \mathbb{N}\backslash[ADG_{min}; ADG_{max\text{-}1}]$

Simplex dominance effect for the alternative allele: $E_{1\text{-}dom\text{-}alt} = |\bar{x}(ADG_{min}) - \bar{x}(ADG_k)|$, where $k$ is $\forall\, x \in \mathbb{N}\backslash[ADG_{min+1}; ADG_{max}]$

$ADG_{min}$ can be nulliplex or simplex, and $ADG_{max}$ can be triplex or quadruplex.

For selected SNPs showing a minimum effect size of 30% (AR formation percentages after 6 weeks), 3.0 roots per rooted cutting, or 0.1 g root fresh mass per rooted cutting, sequences of the SNP´s contigs were used for *BLASTx* analysis at the Genome Database for Rosaceae (https://www.rosaceae.org/blast/nucleotide/protein) [48] using the 'Old Blush' rose genome [30] sequence or via *BLASTx* analysis offered via the National Center for Biotechnology Information (https://blast.ncbi.nlm.nih.gov/Blast.cgi) to identify predicted gene homologues. To achieve the main objective of developing a selection marker for AR formation in rose, we defined the mentioned threshold values for the effect size for the different phenotypic data. These thresholds were set in a manner that allowed focusing on the SNPs with the most substantial effects within the identified peaks. Thereby, SNPs were selected for further investigation, aligning with the main aim of the study.

## DNA extraction

DNA was extracted from young, unfolded leaves stored at room temperature under dark and humid conditions for at least one day to degrade polyphenols and saccharides and subsequently dried over silica gel. The DNA was isolated using the Mag-Bind® M1128 Plant DNA Plus Kit from Omega Bio-tek, Inc. (Norcross, USA). Deviating from the manufacturer's protocol, Phoenix Pure 96 (Hangzhou Allsheng Instruments Co., Ltd., Hangzhou, China) was used instead of the magnetic separation device. In addition, the incubation time was increased to 40 min after the addition of the CSPL buffer, and a previous sorbitol wash step was performed according to [49] to increase the purity of the extracted DNA.

## KASP assay development and validation

Two allele-specific forward primers and one reverse primer were designed for SNP RhK5_69_1627P via 3CR Bioscience´s (Harlow, United Kingdom) free assay design services (https://3crbio.com/free-assay-design/) to perform a KASP assay. For primer development, SNP sequences +/- 50 bases were used. KASP assays were performed following the 3CR Bioscience manual instructions using the PACE® 2.0 Genotyping Master Mix (https://3crbio.com/wp-content/uploads/2022/06/PACE-2.0-User-Guide-v1.1.pdf). For garden roses, the KASP analysis was run in a final volume of 8 µL using 4 µL PACE® 2.0 Genotyping Master Mix, 0.12 µL primer mix (100 µM), 3.88 µL DNAse free water, and 10 ng DNA. For the cut roses and the independent validation set, the KASP was run in a total volume of 5 µL consisting of

2.5 μL PACE® 2.0 Genotyping Master Mix, 0.069 μL primer mix (100 μM), 2.431 μL DNAse free water, and 15 ng DNA. KASP thermocycling and genotyping were performed using a QuantStudio™ 6 Flex Real-Time PCR system (Applied Biosystems, Waltham, USA) applying a standard protocol: 15 min activation at 94°C, followed by ten cycles at 94°C for 20 s and 1 min at 65 to 57°C (decreasing by 0.8°C per cycle), followed by 32 cycles at 94°C for 20 s and 55°C for 1 min, and ending with one final postread stage at 30°C for 30 s.

## Statistical analyses

All statistical analyses were performed using $R$ software [41]. Using the $R$ package *lme4* [50], phenotypic data were analysed in (generalised) linear mixed models to test for significant differences. To test for differences between experiments, (generalised) linear mixed models with the experiment as a fixed effect and genotype and block as random effects were used, including only the 13 garden rose genotypes, which were also tested as references in CR1 and CR2. For differences between genotypes and panels, data were analysed separately in (generalised) linear mixed models with genotype or panel as a fixed effect and experiment and block as random effects. For binomial rooting data after 3 to 6 weeks, generalised linear mixed models were used assuming a binomial distribution, whereas root number per rooted cutting was analysed in a generalised linear mixed model under the assumption of a negative binomial distribution, and root fresh mass per rooted cutting was analysed as ordered quantile normalising transformed data within linear mixed models. The best fitting transformation was identified via the $R$ package *bestNormalize* [51]. Fixed factor effects were analysed via deviance analyses (binomial and negative binomial distributed data) or analysis of variance (metric data), followed by pairwise comparisons (Tukey, $p<0.05$) in case of significant effects of the fixed factor experiment using the $R$ package *emmeans* [52].

To test for significant differences in mean phenotypic values between genotypes of different ADGs for one SNP, the nonparametric Kruskal–Wallis test of the $R$ package *agricolae* was employed [53] followed by Fisher's LSD post hoc test considering Holm–Bonferroni adjustment.

Pearson´s correlation coefficients for all described AR formation traits were calculated using the $R$ package *stats* [41] and visualised in a correlation matrix plot using the $R$ package *corrplot* [54]. Further graphs were created using the $R$ package *ggplot2* [55].

## Results

### AR formation traits

**AR formation percentages.**  All experiments to phenotype AR formation were performed in a hydroponic rooting system under greenhouse conditions (S1 Fig). This enabled evaluation of root formation over time and an easier final evaluation, since no substrate residues had to be removed from developing ARs. The garden rose 'Frühlingsduft' (FD) was excluded from the analyses due to a low number (n<3) of blocks with surviving cuttings. The mean rooting percentages for the remaining 190 genotypes ranged from 0 to 58.8% in week 3 (Fig 1A), 0 to 94.1% in week 4 (Fig 1B), and 0 to 100% in weeks 5 and 6 (Fig 1C and 1D). After 6 weeks, 18 genotypes (14 garden roses, four cut roses) did not show any rooting, while eleven genotypes (seven garden roses, four cut roses) showed rooting in 100% of the surviving cuttings (Fig 1D). Statistical analyses revealed significant differences between genotypes from week 5 on (ST4A in S4 Table). In weeks 4 and 5, differences between experiments were significant (ST4A in S4 Table). Experiment CR1, which was conducted early in the year (16/03/2021-27/04/2021), differed significantly from experiments GR2 and CR2 (after 4 weeks) or only CR2 (after 5 weeks) (ST4B in S4 Table). Furthermore, genotypes of the panel CR showed higher rooting than those of the panel GR after 3, 4, and 5 weeks, respectively (ST4C in S4 Table).

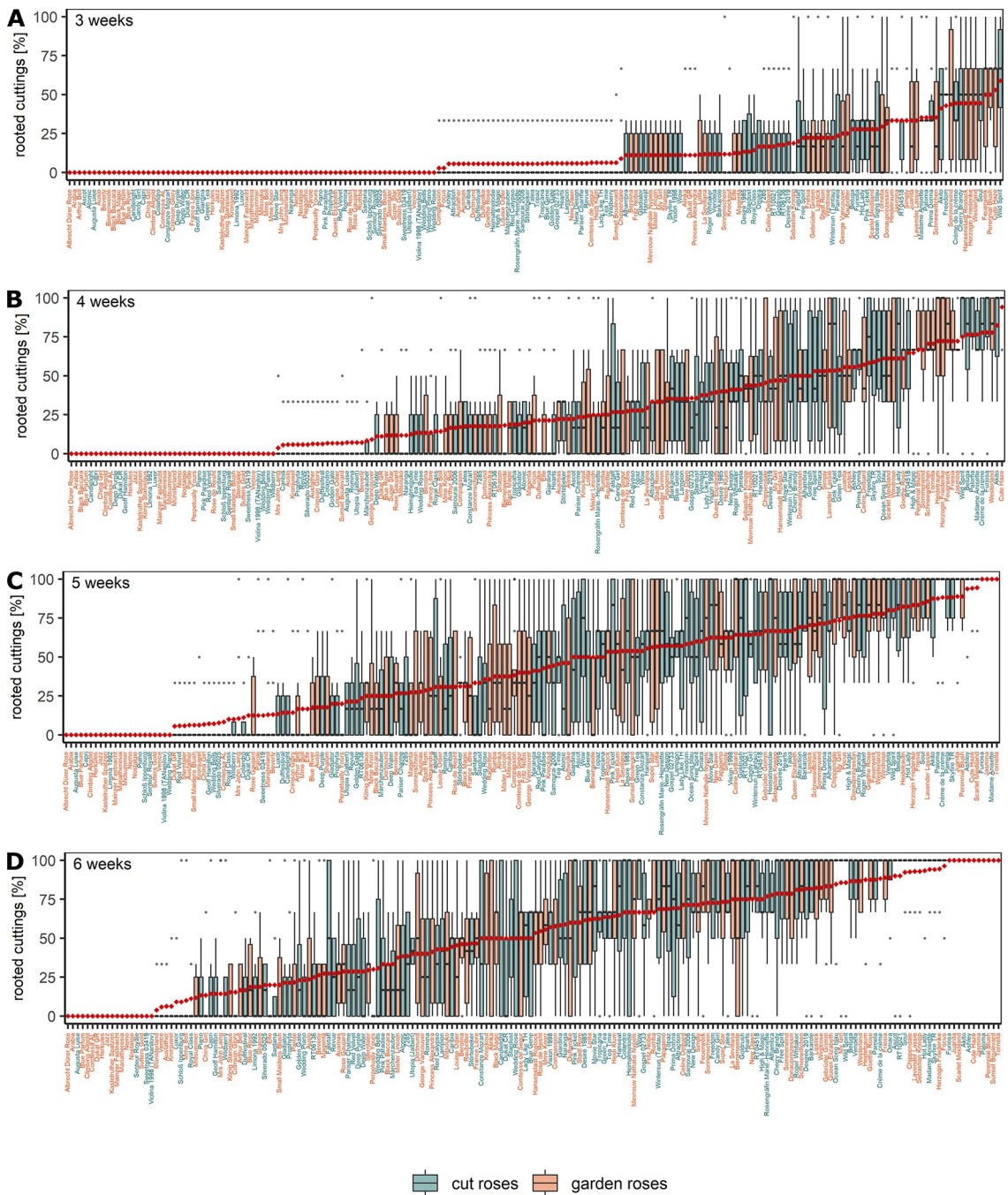

**Fig 1.** Rooting success of 95 garden and 95 cut rose genotypes expressed as the percentage of rooting on surviving cuttings after 3 (A), 4 (B), 5 (C), and 6 weeks (D) of cultivation in a hydroponic system. Data are presented as weighted means per block per separate experiment. Genotypes are ordered based on their weighted overall means (◆). Replicate numbers and cuttings per replicate are given in ST5A-ST5D in S5 Table.

## Root number and root fresh mass

In addition to rooting percentages, the magnitude of AR formation was recorded in terms of root numbers and root fresh masses. Eighteen of the 190 tested genotypes did not form any ARs, and 28 genotypes did form ARs on only one or two cuttings. These 46 genotypes with AR formation

on less than three cuttings were excluded from the statistical analyses to allow for meaningful comparisons. Mean root number ranged from 1.4 (garden rose 'Princess Alexandra') to 22.2 roots per rooted cutting (garden rose 'Celine Delbard') (Fig 2A), whereas the mean root fresh mass varied between 0.025 (garden rose 'Beverly') and 1.514 g (garden rose 'Westerland') (Fig 2B). For both root number and root fresh mass per rooted cutting, the factor experimental repetition was not significant, but the factor genotype had a significant effect (ST4A in S4 Table).

## Correlations between AR formation traits

Pearson's correlation coefficients calculated for rooting percentages after 3 to 6 weeks as well as root numbers and total root fresh masses were all positive and highly significant (Fig 3,

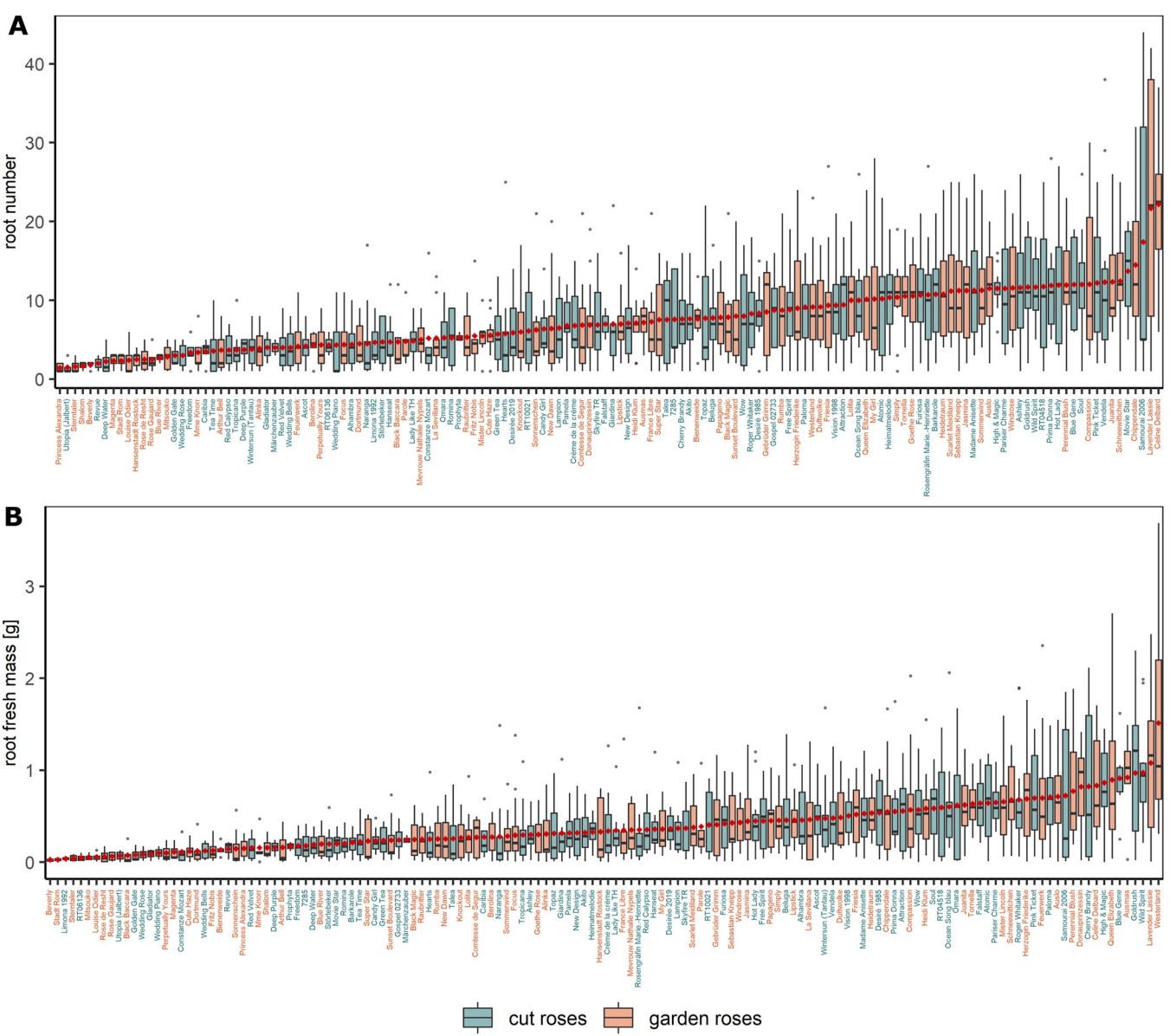

**Fig 2.** Root number (A) and root fresh mass (B) per rooted cutting for 76 cut and 68 garden roses after 6 weeks of cultivation in a hydroponic system. Data are presented as the means per block per separate experiment. Genotypes are ordered based on their weighted overall means (◆). Replicate numbers and cuttings per replicate are given in ST5D in S5 Table.

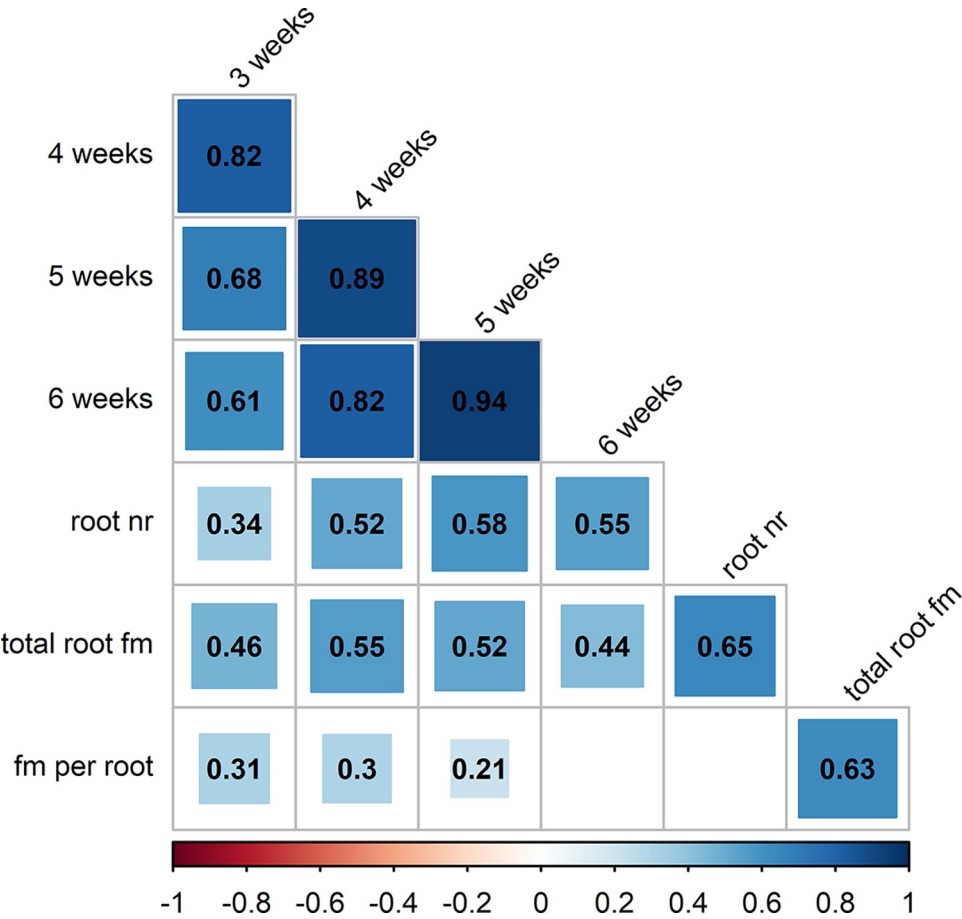

**Fig 3. Pearson´s correlation coefficient matrix for AR formation traits of 190 or 144 genotypes.** Pearson´s correlation coefficients between AR formation percentages after 3, 4, 5, and 6 weeks, root number (root nr) and total root fresh mass (total root fm) per rooted cutting, as well as calculated average fresh mass per root (fm per root) for 190 genotypes (correlations between AR formation %) or 144 genotypes (correlations involving root number and root fresh mass). Values in squares show the significant ($p<0.05$) correlation coefficients, with darker colours and larger squares visualising higher coefficients. *P* values for correlations are shown in ST4D in S4 Table.

ST4D in S4 Table). The highest correlation coefficients of 0.61 to 0.94 were noticed for rooting percentages at the different time points with decreasing correlation coefficients at increasing intervals between the evaluations. The strongest correlation was found between rooting percentages after 5 and 6 weeks (0.94) (Fig 3). Root number and total root fresh mass per rooted cutting were correlated with a coefficient of 0.65 (Fig 3). The calculated average fresh mass per root was significantly correlated with rooting percentages after 3 (0.31), 4 (0.3), and 5 weeks (0.21) in a decreasing manner, but not with rooting percentages after 6 weeks (Fig 3). The correlation with total root fresh mass per rooted cutting was high (0.63) (Fig 3).

## Marker–trait association analyses

**AR formation percentages.** Association mapping for AR formation percentages after 3 to 6 weeks revealed only one significantly associated SNP considering the adjusted *p* value threshold of $4.5*10^{-6}$ ($-\log_{10}(p) = 5.35$): SNP RhK5_4872_1159Q (Chr00, 24.060.509 bp) with a *p* value of $2.19*10^{-6}$ ($-\log_{10}(p) = 5.64$) (Fig 4). However, some of the ADGs were underrepresented (S2 Fig). Therefore, it was only meaningful to include duplex to quadruplex ADGs in

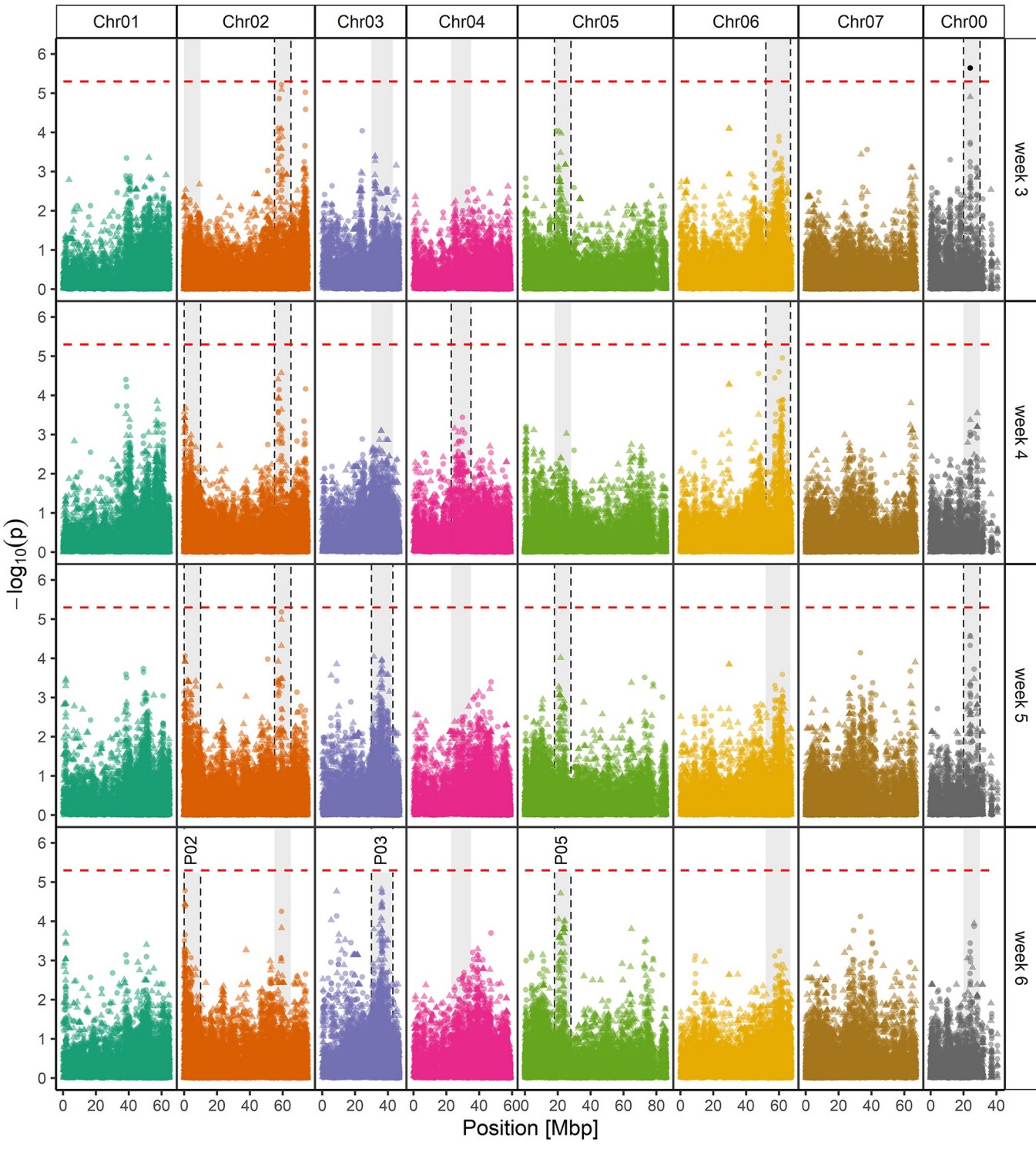

**Fig 4.** Manhattan plots for the association of AR formation percentages after 3 (A), 4 (B), 5 (C), and 6 weeks (D) of cultivation in a hydroponic system. The results of marker trait associations of 25,333 SNPs and AR formation percentages were analysed with an additive model ($\bullet$) or a simplex dominance model ($\blacktriangle$) and shown as $-\log_{10}$ of the SNP´s specific $p$ value. The x-axis shows the positions with respect to the seven *Rosa chinensis* chromosomes [30] (Chr01-Chr07) in megabase pairs (Mbp). Chr00 covers contigs with SNPs that have not yet been mapped. The horizontal dashed red line indicates the *M.eff* corrected $p$ value significance threshold of 5.35 ($-\log_{10}(4.5*10^{-6})$). Distinct peaks for data of at least one evaluation time point are highlighted with grey backgrounds, whereas

this background was framed by black dashed lines when the peak was detected for the respective time point. P02, P03 and P05 after 6 weeks of rooting were analysed in more detail.

the statistical effect size calculations. After 3 and 6 weeks, significantly lower AR formation percentages for quadruplex genotypes compared to duplex and triplex ADGs could be observed (S2 Fig). However, even the highest effect in AR formation after 6 weeks observed between ADGs 2 and 4 was comparatively low at 24.1%. *BLASTx* revealed the SNP RhK5_4872_1159Q contig sequence to be derived from a coding sequence for *Ca²⁺-activated rectifying K⁺ channel 6* (GDR) or *two pore potassium channel 3* (NCBI). Therefore, we included distinctive peaks separated from background noise in our downstream analyses. The peak on Chr04 (at 23–35 Mbp) only appeared for rooting after 4 weeks of AR formation, whereas peaks on Chr02 (at 55–65 Mbp), Chr06 (at 52–67 Mbp), and Chr00 (at 20–30 Mbp) were observed for data collected after 3, 4, and 5 weeks, 3 and 4 weeks, or 3 and 5 weeks of rooting, respectively.

In the following, we only focused on peaks that were detected at the time of the final evaluation after 6 weeks. These are indicated in Fig 4 as peaks P02 (Chr02, 0–10 Mbp, also present after 4 and 5 weeks), P03 (Chr03, 30–43 Mbp, also present after 5 weeks) and P05 (Chr05, 18–28 Mbp, also present after 3 and 5 weeks), and were the peaks closest to the *p*-value threshold. In-depth analyses of P02, P03, and P05 revealed 40 SNPs showing significant allele dosage effects (Kruskal–Wallis and LSD post hoc *p*<0.05) on AR formation percentages with detected effect sizes of 30% to 40% (ST6A in S6 Table). Moreover, 14 SNPs showed an effect size of 40% or more, of which twelve were located in P02 (Chr02, 0–10 Mbp) and two SNPs were in P05 (Chr05, 18–28 Mbp) (Table 1). Nine out of the ten SNPs with the highest effect sizes on AR formation, ranging between 41.6% (RhK5_69_3887Q) and 47.8% (RhK5_69_3401P), were located within the same contig RhK5_69, representing a segment of the coding region for a *sacI homology domain-containing protein/WW domain-containing protein* (BLASTx GDR) or more precisely the *probable phosphoinositide phosphatase SAC9* (BLASTx NCBI) (Table 1). In addition, the contig sequence of SNP Rh12GR_145_4298Q (effect size of 40.5%) could be mapped to the identical coding region in the same genomic position. The allele configuration of homozygous genotypes for the nine SNPs from contig RhK5_69 and the SNP Rh12GR_145_4298Q revealed collinearity regarding the allele dosages for the 10 SNPs within *SAC9* (S7 Table).

The two additional SNPs in peak P02 are located in a *pyridine nucleotide-disulfide oxidoreductase* (GDR) or *NADPH:adrenodoxin oxidoreductase* (NCBI) (RhK5_6822_287Q, 43.4%) and in a *nudix hydrolase homolog 20* (Rh12GR_52692_217P, 40.1%), respectively (Table 1). The two SNPs in P05 were in a region coding for *transcription factor GTE4* (RhK5_11050_203Q) and a *P-loop containing nucleoside triphosphate hydrolase superfamily protein* (RhK5_6493_408P, GDR) (Table 1). Interestingly, the *BLASTx* results for the SNPs showing effects between 30 and 40% identified one further SNP, namely, RhK5_4257_1151Q (38.1%), to be located in the same coding region for *nudix hydrolase homolog 20* in P02. Two additional SNPs in the same contig, RhK_570_626P (35.2%) and RhK5_570_439P (34.8%), are located within the coding region for *nudix hydrolase homolog 19* (ST6A in S6 Table). Several genes previously reported to be related to AR formation and reviewed by Guan et al. [56], Druege et al. [10], or Li [57] colocalised with the identified peaks (S8 Table), for example *ACO1* (*ACC oxidase 1*), *HXK* (*hexokinase*), *CDC27* (*cell division cycle protein 27*), *YUC* (*indole-3-pyruvate monooxygenase YUCCA*), or *CYCA1* (*CYCLIN A1*) in peak P02, *AUX1* (*auxin transporter protein 1*) and *SCR* (*scarecrow-like protein*) in peak P03 and *NAC1* (*NAC domain containing protein 1*), *TPS1* (*trehalose-6-phosphate synthase 1*), and *RPD1* (*root primordium defective 1*) in peak P05 (S8 Table).

**Table 1. Selected SNP markers located in distinct peaks in Manhattan plots for AR formation percentages after 6 weeks with minimal effect sizes of 40%.** SNP markers are ordered by their effect size (rooting %), specified as additive effect unless otherwise stated. Corresponding gene predictions are based on *BLASTx* searches on GDR and NCBI.

| Marker | Peak | Chr | Position | Effect [%] | Gene prediction |
|---|---|---|---|---|---|
| RhK5_69_3401P | P02 | Chr02 | 693852 | 47.8 | GDR: sacI homology domain-containing protein / WW domain-containing protein<br>NCBI: probable phosphoinositide phosphatase SAC9 [*Rosa chinensis*] |
| RhK5_69_2539P | P02 | Chr02 | 691664 | 46.5 | GDR: sacI homology domain-containing protein / WW domain-containing protein<br>NCBI: probable phosphoinositide phosphatase SAC9 [*Rosa chinensis*] |
| RhK5_69_728Q | P02 | Chr02 | 688720 | 46.5 | GDR: sacI homology domain-containing protein / WW domain-containing protein<br>NCBI: probable phosphoinositide phosphatase SAC9 [*Rosa chinensis*] |
| RhK5_69_3284P | P02 | Chr02 | 693735 | 46.5 | GDR: sacI homology domain-containing protein / WW domain-containing protein<br>NCBI: probable phosphoinositide phosphatase SAC9 [*Rosa chinensis*] |
| RhK5_69_3362P | P02 | Chr02 | 693813 | 46.5 | GDR: sacI homology domain-containing protein / WW domain-containing protein<br>NCBI: probable phosphoinositide phosphatase SAC9 [*Rosa chinensis*] |
| RhK5_69_2848P | P02 | Chr02 | 692219 | 44.7 | GDR: sacI homology domain-containing protein / WW domain-containing protein<br>NCBI: probable phosphoinositide phosphatase SAC9 [*Rosa chinensis*] |
| RhK5_69_1627P | P02 | Chr02 | 690585 | 44.7 | GDR: sacI homology domain-containing protein / WW domain-containing protein<br>NCBI: probable phosphoinositide phosphatase SAC9 [*Rosa chinensis*] |
| RhK5_6822_287Q | P02 | Chr02 | 4796243 | 43.4 | GDR: Pyridine nucleotide-disulphide oxidoreductase family protein<br>NCBI: NADPH:adrenodoxin oxidoreductase, mitochondrial isoform X1 [*Rosa chinensis*] |
| RhK5_69_3990P | P02 | Chr02 | 694441 | 42.7 | GDR: sacI homology domain-containing protein / WW domain-containing protein<br>NCBI: probable phosphoinositide phosphatase SAC9 [*Rosa chinensis*] |
| RhK5_69_3887Q | P02 | Chr02 | 694338 | 41.6 | GDR: sacI homology domain-containing protein / WW domain-containing protein<br>NCBI: probable phosphoinositide phosphatase SAC9 [*Rosa chinensis*] |
| RhK5_11050_203Q | P05 | Chr05 | 24053773 | 41.4 | GDR: global transcription factor group E4<br>NCBI: transcription factor GTE4 isoform X1 [*Rosa chinensis*] |
| RhK5_6493_408P | P05 | Chr05 | 24078876 | 41.2 | GDR: P-loop containing nucleoside triphosphate hydrolases superfamily protein<br>NCBI: protein SEEDLING PLASTID DEVELOPMENT 1 [*Rosa chinensis*] |
| Rh12GR_145_4298Q | P02 | Chr02 | 694565 | 40.5 | GDR: sacI homology domain-containing protein / WW domain-containing protein<br>NCBI: probable phosphoinositide phosphatase SAC9 [*Rosa chinensis*] |
| Rh12GR_52692_217P | P02 | Chr02 | 4652672 | 40.1 | GDR: nudix hydrolase homolog 20<br>NCBI: nudix hydrolase 20, chloroplastic [*Rosa chinensis*] |

**Root number and root fresh mass.** Association mapping for root number revealed no significantly associated SNPs but three distinctive peaks that were further analysed, namely, P04 (Chr04, 23–35 Mbp), P06nr (Chr06, 38–52 Mbp), and P07 (Chr07, 20–35 Mbp) (Fig 5). P04 was also found for rooting percentages after 4 weeks (Fig 4). Ten SNPs showed a significant allele dosage effect (Kruskal–Wallis and LSD post hoc $p < 0.05$) of more than three roots per rooted cutting, five SNPs in P07, four in P06nr, and one in P04 (Table 2). Three of the five SNPs with the highest additive effects of 5.2 (RhK5_19325_1901P, *plastidial pyruvate kinase*), 4.6 (Rh12GR_13953_739P, NA) and 4.0 roots per rooted cutting (RhK5_3575_1325Q, *GRAS family transcription factor* (GDR) or *scarecrow-like protein 6* (NCBI)) were located in P07 (Chr07, 20–35 Mbp) (Table 2). Three SNPs in P06nr showed an effect of 4.9 (RhK5_10139_474Q, protein of unknown function), 4.4 (RhMCRND_28067_505P, *pleiotropic drug resistance protein*), and 3.9 roots (RhK_15295_125P, *RING/U-box superfamily protein* or *nucleoporin NUP159*) (Table 2).

Association mapping for root fresh mass revealed no significantly associated SNPs, but four distinctive peaks were further analysed: P03fm (Chr03, 30–43 Mbp), P04 (Chr04, 23–35 Mbp), P06fm (Chr06, 52–67 Mbp), and P07 (Chr07, 20–35 Mbp) (Fig 5). The position of P04 was identical to the position of peaks detected for root number and AR formation percentages after 4 weeks and the position of P07 with the peak detected for root number only (Figs 4 and 5). Analysing the peaks P03, P04, P06fm, and P07 for root fresh mass per rooted cutting in

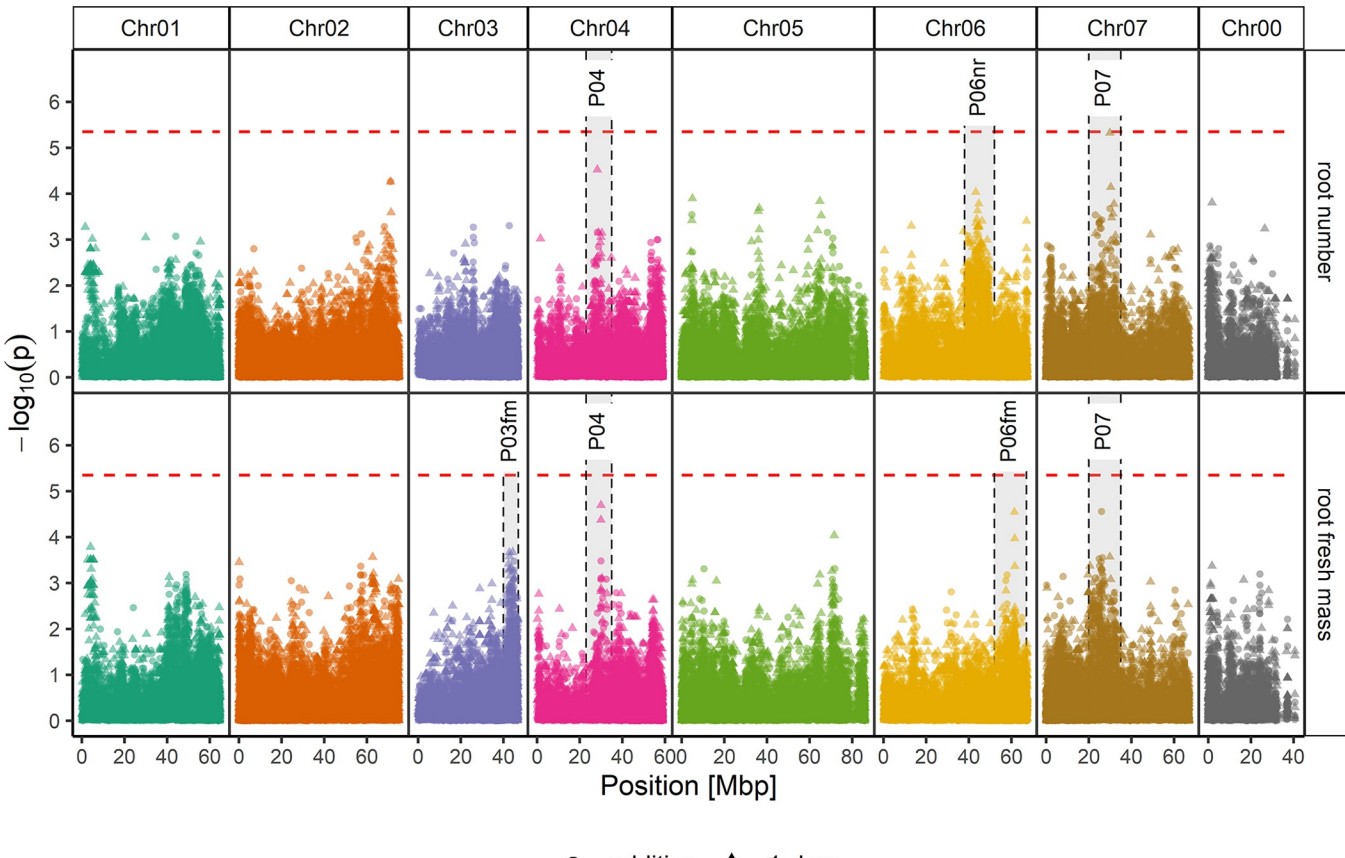

**Fig 5.** Manhattan plots for the association of root number (A) and root fresh mass (B) per rooted cutting after 6 weeks of cultivation in a hydroponic system. The results of marker trait associations of 25,333 SNPs and AR traits root number (A) and root fresh mass (B) per rooted cutting were analysed with an additive model (•) or a simplex dominance model (▲), shown as–$\log_{10}$ of the SNP´s specific $p$ value. The x-axis shows the positions with respect to the seven *Rosa chinensis* chromosomes [30] (Chr01-Chr07) in megabase pairs (Mbp). Chr00 covers contigs for which positions have not yet been mapped. The horizontal dashed red line indicates the *M.eff* corrected $p$ value significance threshold of 5.35 (-$\log_{10}(4.5*10^{-6})$). Distinct peaks that were analysed in detail are highlighted with grey backgrounds and framed by black dashed lines.

more detail revealed 15 SNPs that fulfilled our defined selection criteria and showed significant effects of more than 0.1 g root fresh mass per rooted cutting (Table 3). Ten of these were located in P03fm, three SNPs in P06fm, and two SNPs in P07 (Table 3). Two of the three SNPs with the highest additive effects were located on Chr07: RhK5_19325_1901P (*plastidial pyruvate kinase*), with an effect of 0.41 g fresh mass, and Rh12GR_13953_739P, with an effect of 0.29 g (Table 3). These two SNPs were also shown to have significant effects on root number (Table 2). In addition, effects for SNPs located in coding regions of putative candidate genes could be found for RhMCRND_23321_187P (*WD40 repeat-like* protein, 0.22 g) and RhMCRND_760_1054Q (*leucine-rich repeat protein kinase family* protein, 0.19 g) (Table 3). Rose homologues of genes that had already been described and reviewed in the literature by Guan et al. [56], Druege et al. [10], or Li [57] to be involved in AR formation and positioned in peaks detected for root number and root fresh mass per rooted cutting (S8 Table) were *ERF1* (*ethylene response factor 1*) and *BBM* (*AP2-like ethylene-responsive transcription factor BBM*) in P03fm, eight gene homologues (*YUC, LBD29, CYCB2, AtD27, HDA19, LBD16, IAA14, RID2*) in P06nr and 13 gene homologues (*LAX3, YUC, MdTIR1, CYCA1, CYCA3, MAX4/ CCD8, MAX1, HXK3, TPP, YUC1, SUC, ANT, TPL*) in P06fm. In P07, six AR-related genes

**Table 2. Selected SNP markers located in distinct peaks in Manhattan plots for root number after 6 weeks with a minimal effect size of 3 roots per rooted cutting.**
SNP markers are ordered by effect size, specified as additive effect unless otherwise stated. Corresponding gene predictions are based on *BLASTx* searches on GDR and NCBI. *NA* indicates that no gene of known function could be assigned.

| Marker | Peak | Chr | Position | Effect [root number] | Gene prediction |
|---|---|---|---|---|---|
| RhK5_19325_1901P | P07 | Chr07 | 25973632 | 5.4 | GDR: plastidial pyruvate kinase 3<br>NCBI: pyruvate kinase isozyme G, chloroplastic [*Rosa chinensis*] |
| RhK5_10139_474Q | P06nr | Chr06 | 45477996 | 4.9 | GDR: Protein of unknown function<br>NCBI: uncharacterized protein LOC112172111 [*Rosa chinensis*] |
| Rh12GR_13953_739P | P07 | Chr07 | 23077698 | 4.6 | GDR: NA<br>NCBI: NA |
| RhMCRND_28067_505P | P06nr | Chr06 | 42341228 | 4.4 | GDR: pleiotropic drug resistance 6<br>NCBI: pleiotropic drug resistance protein 2 [*Rosa chinensis*] |
| RhK5_3575_1325Q | P07 | Chr07 | 20541809 | 4.0 | GDR: GRAS family transcription factor<br>NCBI: scarecrow-like protein 6 [*Rosa chinensis*] |
| RhK5_15295_125P | P06nr | Chr06 | 45110951 | 3.9 | GDR: RING/U-box superfamily protein<br>NCBI: nucleoporin NUP159 isoform X2 [*Rosa chinensis*] |
| RhMCRND_10282_1041P | P07 | Chr07 | 28164800 | 3.7 (1-dom-alt) | GDR: damaged DNA binding<br>NCBI: uncharacterized protein LOC112179222 [*Rosa chinensis*] |
| RhK5_11841_453Q | Chr07 | Chr07 | 20809726 | 3.4 | GDR: histidine triad nucleotide-binding 4<br>NCBI: bifunctional adenosine 5'-phosphosulfate phosphorylase/adenylylsulfatase HINT4 [*Rosa chinensis*] |
| RhMCRND_2021_462Q | P06nr | Chr06 | 45483087 | 3.3 (1-dom-alt) | GDR: NA<br>NCBI: NA |
| RhK5_20722_584P | P04 | Chr04 | 28271196 | 3.1 (1-dom-alt) | GDR: Pentatricopeptide repeat (PPR) superfamily protein<br>NCBI: uncharacterized protein LOC112196367 [*Rosa chinensis*] |

with shared traits of root number and root fresh mass (*ERF1*, *SAUR*, *OsCRL*, *TPP*, *RID*, *HDA19*) were found (S8 Table).

## AR formation traits depending on the allele dosages of Rhk5_69_1627P

The mean effect of the eight SNPs located in contig RhK5_69 on AR formation percentages after a rooting period of 6 weeks was 45.1% (Table 1). SNP RhK5_69_1627P was selected for further analyses of allele dosage effects. At each time point, the quadruplex ADG displayed the maximum mean AR formation percentages, namely, 22.7% after 3 weeks (Fig 6A), 50.4% after 4 weeks (Fig 6B), 68.8% after 5 weeks (Fig 6C), and 77.5% after 6 weeks (Fig 6D), which were significantly higher than those of the four other ADGs, with the exception of the nulliplex ADG after 3 weeks (Fig 6A–6D). For root fresh mass, no significant difference between any of the ADGs was observed (Fig 6F), but a significantly lower root number was recorded for the nulliplex ADG compared to the quadruplex ADG (difference: 4.5 roots) (Fig 6E).

A KASP marker was developed from the sequence of the SNP RhK5_69_1627P (S9 Table), which allowed us to clearly discriminate between all ADGs (S3 Fig). For garden roses, five genotypes could not be clearly assigned to a distinct heterozygous ADG by using fitTetra [43] and therefore were indicated as *NA* (S3 Fig, ST1A in S1 Table). Comparing the allele calling of the homozygous genotypes between the Axiom 68 K array and the KASP marker for SNP RhK5_69_1627P revealed an overall high agreement with the exception of the two genotypes 'Piano' (cut rose) and 'Perennial Blush' (garden rose) (ST1B in S1 Table).

## Validation of the KASP marker RhK5_69_1627P

The KASP marker for SNP RhK5_69_1627P was analysed in an independent set of 377 genotypes (S2 Table), in which 41 nulliplex and 44 quadruplex genotypes were identified, whereas

**Table 3. Selected SNP markers located in distinct peaks in Manhattan plots for root fresh mass after 6 weeks with a minimal effect size of 0.1 g root fresh mass per rooted cutting.** SNP markers are ordered by effect size, specified as additive effect unless otherwise stated. Corresponding gene predictions are based on *BLASTx* searches on GDR and NCBI. *NA* indicates that no gene of known function could be assigned.

| Marker | peak | Chr | Position | Effect [g] | Gene prediction |
|---|---|---|---|---|---|
| RhK5_19325_1901P | P07 | Chr07 | 25973632 | 0.41 | GDR: plastidial pyruvate kinase 3<br>NCBI: pyruvate kinase isozyme G, chloroplastic [*Rosa chinensis*] |
| Rh12GR_50601_161Q | P03fm | Chr03 | 33073922 | 0.29 (1-dom-ref) | GDR: farnesylated protein-converting enzyme 2<br>NCBI: CAAX prenyl protease 2 [*Rosa chinensis*] |
| Rh12GR_13953_739P | P07 | Chr07 | 23077698 | 0.29 | GDR: NA<br>NCBI: NA |
| RhMCRND_24481_461P | P03fm | Chr03 | 33414488 | 0.26 (1-dom-ref) | GDR: Protein kinase superfamily protein<br>NCBI: probable serine/threonine-protein kinase PBL7 isoform X1 [*Rosa chinensis*] |
| Rh12GR_15113_134Q | P03fm | Chr03 | 34213536 | 0.25 (1-dom-ref) | GDR: Major facilitator superfamily protein<br>NCBI: UNC93-like protein 3 [*Rosa chinensis*] |
| RhK5_3480_821Q | P06fm | Chr06 | 57355761 | 0.21 (1-dom-alt) | GDR: pfkB-like carbohydrate kinase family protein<br>NCBI: uncharacterized sugar kinase slr0537 [*Rosa chinensis*] |
| RhK5_8613_484Q | P03fm | Chr03 | 34128560 | 0.21 | GDR: NA<br>NCBI: NA |
| RhMCRND_23321_187P | P03fm | Chr03 | 34486135 | 0.22 (1-dom-ref) | GDR: Transducin/WD40 repeat-like superfamily protein<br>NCBI: WD repeat-containing protein 44 [*Rosa chinensis*] |
| RhK5_2054_843P | P06fm | Chr06 | 61594963 | 0.19 (1-dom-alt) | GDR: OPC-8:0 CoA ligase1<br>NCBI: 4-coumarate—CoA ligase-like 5 [*Rosa chinensis*] |
| Rh12GR_402_674P | P03fm | Chr03 | 31513544 | 0.19 (1-dom-alt) | GDR: NA<br>NCBI: NA |
| RhK5_6551_997P | P03fm | Chr03 | 42763736 | 0.19 (1-dom-alt) | GDR: nuclear factor Y<br>NCBI: nuclear transcription factor Y subunit C-9 [*Rosa chinensis*] |
| RhMCRND_760_1054Q | P03fm | Chr03 | 33215504 | 0.19 (1-dom-ref) | GDR: Leucine-rich repeat protein kinase family protein<br>NCBI: probable LRR receptor-like serine/threonine-protein kinase At1g67720 isoform X2 [*Rosa chinensis*] |
| RhK5_9128_459P | P03fm | Chr03 | 41118695 | 0.15 | GDR: mitochondrially targeted single-stranded DNA binding protein<br>NCBI: single-stranded DNA-binding protein, mitochondrial isoform X1 [*Rosa chinensis*] |
| Rh12GR_12027_800P | P06fm | Chr06 | 61422292 | 0.13 (1-dom-ref) | GDR: NA<br>NCBI: NA |
| Rh12GR_55615_445P | P03fm | Chr03 | 42689624 | 0.12 | GDR: amidase 1<br>NCBI: NA |

only nine of the remaining 292 heterozygous genotypes could not be clearly assigned to one of the three heterozygous ADGs (S2 Table, S4 Fig). Sufficient plant material to test in rooting experiments was available for 15 nulliplex and 28 quadruplex genotypes (S10 Table). Cuttings of three nulliplex genotypes did not survive until the end of evaluation after 6 weeks, which is why they were excluded from the analyses. The phenotyping of AR formation traits showed striking significant allele dosage effects, which ranged from a minimum of 13.5% (3 weeks) to 32.6% (6 weeks), on AR formation percentages for all evaluation time points from 3 to 6 weeks (Fig 7). Rooting percentages for nulliplex genotypes ranged from 0.3% (3 weeks) to 6.5% (6 weeks) and were significantly lower than those recorded for quadruplex genotypes, which ranged from 13.8% (3 weeks) to 39.1% (6 weeks) (Fig 7). Since the number of rooted cuttings was lower than three for all nulliplex genotypes, data for root number and root fresh mass per rooted cutting were not compared between nulliplex and quadruplex ADGs.

## Discussion

We analysed 95 cut and 95 garden roses regarding their ability to form ARs and observed high genotypic variation for AR formation-associated traits. The phenotypic data were used for GWAS approaches. Genomic regions were identified in which several SNPs expressed strong

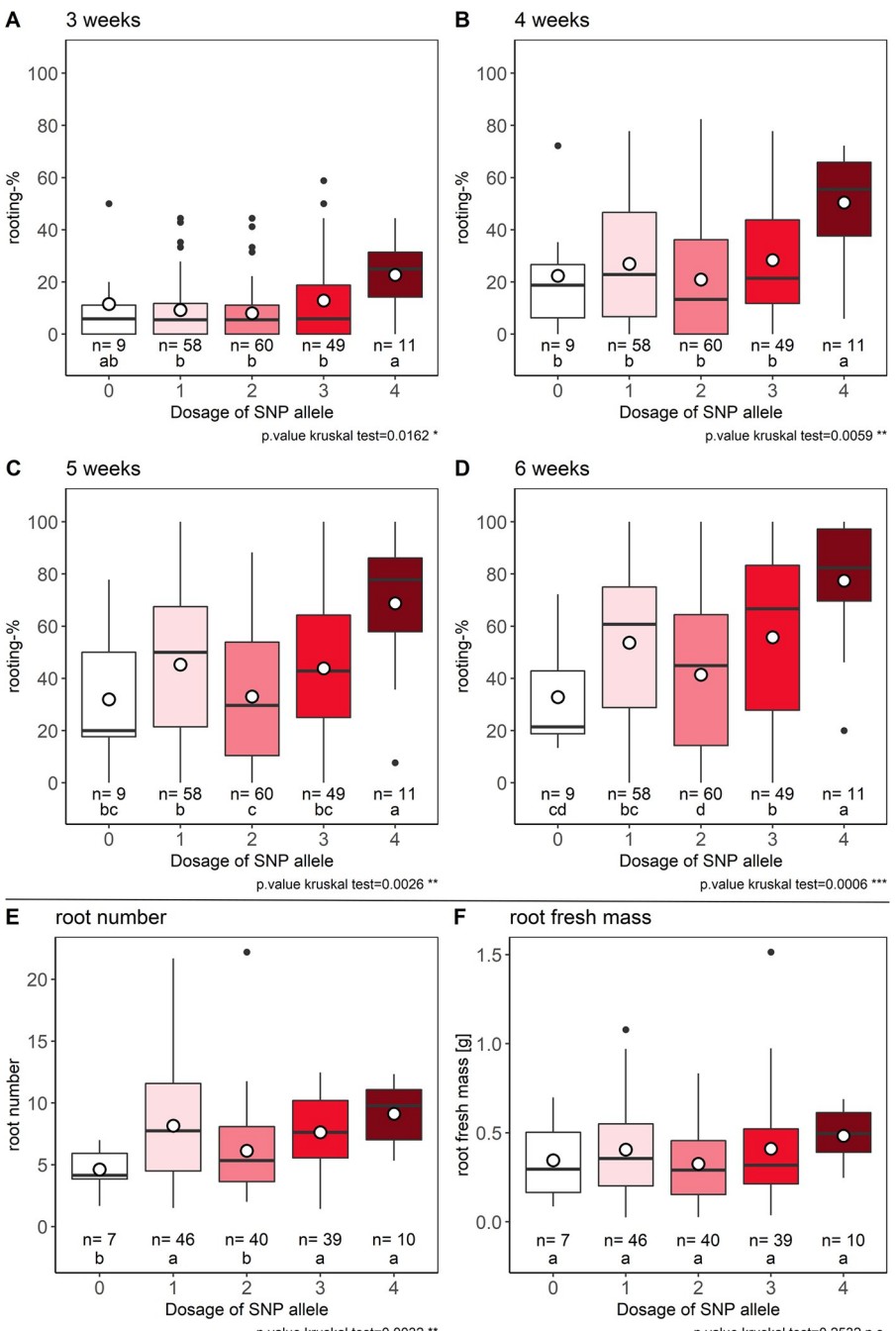

**Fig 6. Effects of allele dosage configuration for SNP RhK5_69_1267P on AR formation traits.** Allele dosage effects for the SNP RhK5_69_1627P on AR formation percentages after 3, 4, 5, and 6 weeks (A-D), root number (E) and root fresh mass per rooted cutting (F). X-axis values show the dosages for the SNP allele from nulliplex (0) to quadruplex (4). The number of individuals per ADG is given by *n*. Letters indicate significance groups as determined by Fisher´s LSD criterion for $p<0.05$ under consideration of the Holm–Bonferroni adjustment separately for each trait.

and significant effects on the traits dependent on their ADG configuration. One SNP was converted into a KASP marker and successfully confirmed in an independent set of homozygous genotypes to show a strong allele-dependent effect on AR formation.

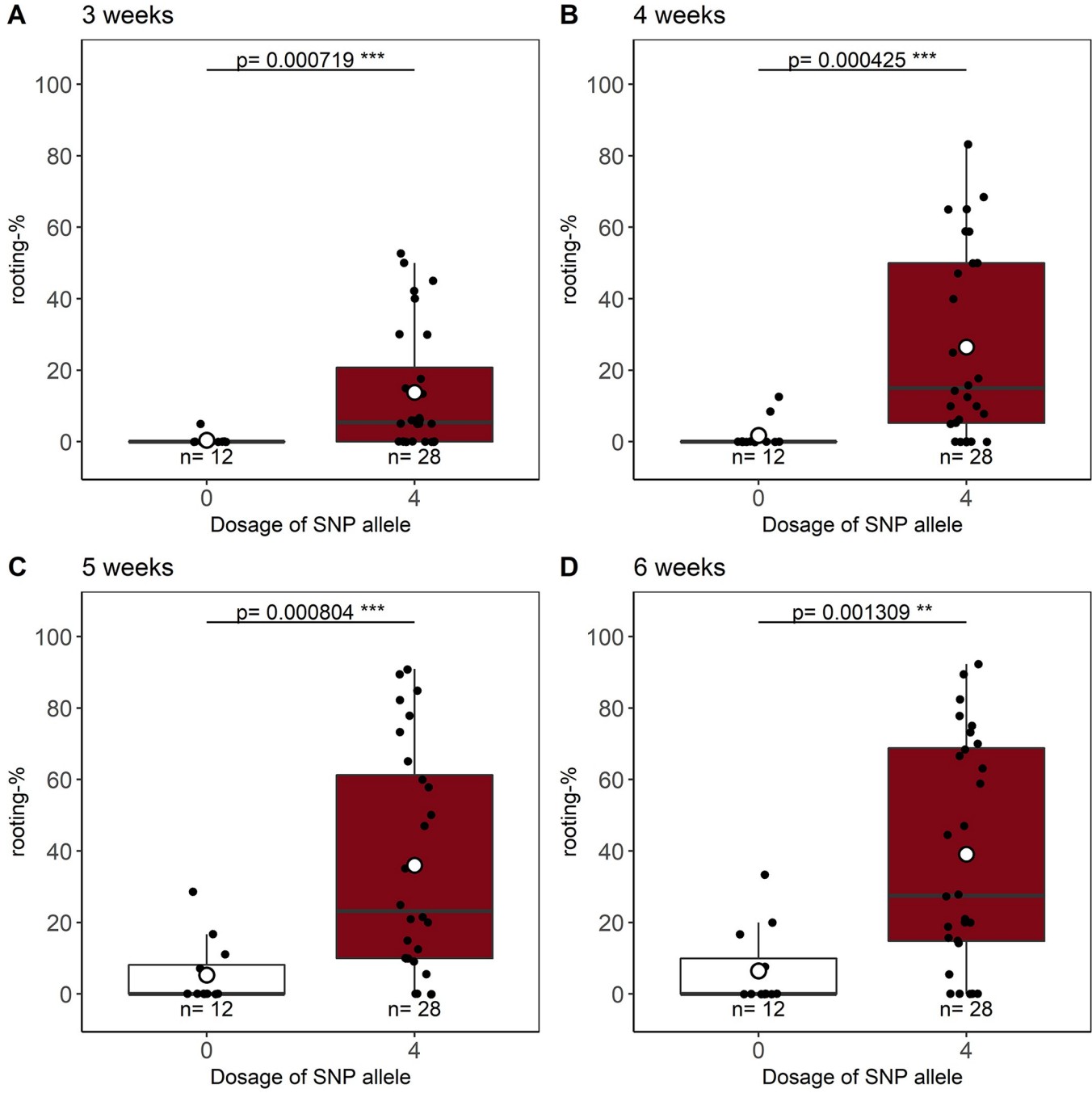

**Fig 7. Effects of allele dosage configuration for SNP RhK5_69_1267P on AR formation percentages for homozygous genotypes selected from an independent population by KASP marker analysis.** Allele dosage effects for the SNP RhK5_69_1627P on AR formation percentages after 3, 4, 5, and 6 weeks (A-D) were analysed for 15 nulliplex and 28 quadruplex genotypes. Homozygous genotypes were selected with the SNP-related KASP marker from an independent population of 377 genotypes. X-axis values show the dosages for the SNP allele, either nulliplex (0) or quadruplex (4), and the number of individuals per ADG is given by $n$. Differences between ADGs are given by $p$ values separately for the evaluation weeks (* $p < 0.05$, ** $p < 0.01$, *** $p < 0.001$).

## Genotypic variation in AR formation

Strong genotypic differences were observed for the 190 rose genotypes with respect to the rooting percentages as well as the root number and root fresh mass per rooted cutting for the 144 root-forming genotypes (Figs 1 and 2). At the final evaluation after 6 weeks, AR formation

ranged from 0 to 100%. This is in close agreement with reports on large genotypic differences in AR formation by Nguyen et al. [7], who tested the same set of garden roses, and Dubois and de Vries [11], who investigated 50 miniature roses. This variation underlines the need to unravel the reasons for such strong genotypic differences in AR formation in rose, as a poor rooting response is the main limiting factor in implementing autovegetative propagation by cuttings for certain roses. At the same time, it indicates that breeding for good rooting ability should be possible.

Significant differences in rooting percentages between experiments were observed after 4 and 5 weeks (S4 Table). These might be due to environmental factors affecting the status of nutrients and carbohydrates in stock plants and cuttings, as described by Druege [58] and Otiende et al. [59]. Likewise, Hambrick et al. [60] showed a seasonal effect on AR formation in field-grown hardwood cuttings of *R. multiflora*. However, as differences were observed only after 4 and 5 weeks, and neither at the time of the final evaluation after 6 weeks nor for root number and root fresh mass, the season seemed to influence the speed of AR formation rather than the quality. In addition, significantly higher rooting percentages in panel CR compared to panel GR after 3, 4, and 5 weeks, but not after 6 weeks and not for root number and root fresh mass per rooted cutting, led to the conclusion, that the genotypes of the two panels differed in rooting speed, but not in rooting quality. However, these differences could be due to seasonal differences, as only some of the garden rose genotypes were tested synchronously with the cut rose genotypes in the same experiments.

Almost all observed AR formation traits were significantly and positively correlated with each other (Fig 3). Only the calculated average fresh mass per root was not significantly correlated with the parameters rooting percentages after 6 weeks and root number. Correlations between different evaluation time points for AR formation percentages decreased with increasing intervals between the evaluation, whereas the highest correlation of 0.94 was observed between values after 5 and 6 weeks (Fig 3). These results and slightly higher correlations between rooting percentages after 5 weeks and root number and root fresh mass indicate that an end evaluation even after 5 weeks of cultivation could be reasonable and shorten the duration of experiments. However, setting earlier time points for the final evaluation could lead to underestimation of the actual ability of genotypes to form ARs, because 42 genotypes did not form any ARs after 4 weeks (Fig 1). The correlations between FM per root and rooting percentages suggest that there may be a link between growth time after root penetration and the qualitative constitution of a single root (Fig 3).

## Marker–trait associations to genetically dissect AR formation

GWAS has become a popular approach in plant sciences to study the genetic regulation of complex and highly quantitative traits such as biotic stress resistance, abiotic stress tolerance, and yield traits, especially in important agricultural species [61]. Moreover, AR formation was also addressed by GWAS in different species, namely, wound-induced AR formation in poplar [62] and rose [7], waterlogging-triggered AR formation in barley [63], wheat [64], and maize [65], or crown root development in maize [66]. Additionally, the GWAS approach has been quite frequently and successfully used to unravel the genetic factors underlying complex traits, particularly for polyploid ornamentals such as chrysanthemums [67, 68], *Phalaenopsis* orchids [69] and especially rose [30, 36–39].

In our study, for the first time, a GWAS approach was followed for rooting percentages over time and for a comparatively high number of 190 genotypes. Overall, only one significantly associated SNP (RhK5_4872_1159Q, Chr00) was detected for rooting percentage after 3 weeks. This SNP is located in the coding region for a $Ca^{2+}$-activated potassium channel, *TPK3/*

*KCO6*. Interestingly, the potassium channel *AKT1* has been shown to be involved in the regulation of auxin-mediated root growth [70]. Considering further the unequal distribution of individuals to the ADGs and the relatively small effect size on the AR formation results after 6 weeks, the SNP was classified as not useful for marker-assisted selection of AR formation (S2 Fig).

In addition to the significant SNP, the GWAS revealed several distinct peaks for different genomic regions at different times of evaluation (Fig 4). Some peaks were recognizable only for earlier time points (Chr02, 55–65 Mbp; Chr06, 52–67 Mbp) or for one week (Chr04, 23–35 Mbp) (Fig 4), whereas other peaks were not distinct at early time points but formed at later time points (Chr02, 0–10 Mbp; Chr03, 30–43 Mbp) (Fig 4). This indicates that different genes may be involved in different phases of AR formation and root growth. Since one major goal of this study was to identify markers for an improved rooting ability, we focused on the final evaluation after 6 weeks of rooting. Three distinct peaks could be observed for this time point: P02 (Chr02, 0–10 Mbp), P03 (Chr03, 30–43 Mbp), and P05 (Chr05, 18–28 Mbp) (Fig 4). P02 matched a peak observed for *in vivo* root dry mass in a previous study [7].

Sequences of genes reported to play a role in AR formation [10, 56, 57] were blasted against the *R. chinensis* 'Old Blush' genome [30] to uncover genes that colocalise with peaks P02, P03 and P05 (S8 Table). Genes involved in AR formation could be assigned to genomic regions of all three peaks: auxin-related genes such as *YUC* and *GNOM*, cell division-associated genes such as *HBT*/*CDC27*, *CYCA1*, and *RID2*, the carbohydrate-associated gene *HXK*, and *ACO1*, which is involved in ethylene synthesis, were located in P02. Within P03, a homologue of the auxin transporter *AUX1* and *SCR*, encoding a transcription factor of the *GRAS* family and involved in auxin signalling and homeostasis with a promoting role in AR formation, were found [58, 71]. P05 covered the cell division-associated gene *RPD1*, the carbohydrate synthase gene *TPS1* and the wounding-induced transcription factor *NAC1* [72].

**SNPs in peaks show prominent allele dosage effects on rooting percentages.**  The SNPs in the peak regions for AR formation percentages after 6 weeks were filtered as explained earlier to reveal 40 SNPs showing notable allele dosage effects of 30 to 40%, of which ten were localised in P02, three in P03, and 27 in P05 (S6A Table). One of these SNPs, namely, RhK5_11361_109P (P03, effect size: 32.9%), is located in a homologue of *CTR1*, which has been described to negatively regulate ethylene signal transduction and to suppress AR formation in *A. thaliana* hypocotyls [73]. Furthermore, SNPs were identified in gene homologues with functions in processes closely related to AR formation: *ROP9* (RhMCRND_2994_1008P, P02, effect 31.6%) negatively regulates auxin-induced gene expression and promotes expression of abscisic-acid induced genes, which were described to inhibit AR formation [74, 75]. *IDM1* (RHK5_14306_391Q, P05, effect size: 31%) was mentioned to prevent DNA (hyper) methylation. DNA methylation critically influences AR formation ability, especially in adult plants [76]. Very conspicuous were four SNPs with effect sizes of 40.1% (Rh12GR_52692_217P), 38.1% (RhK5_4257_1151Q), 35.2% (RhK5_570_626P), or 34.8% (RhK5_570_439P), all localising in P02 in gene homologues for *nudix hydrolase 19* or *nudix hydrolase 20*, respectively. *Nudix hydrolase 19* has been described to be involved in NADP(H)-mediated redox homeostasis, and loss-of-function mutants showed higher tolerance to abiotic stresses such as high light conditions or arsenic-induced stress [77, 78].

In total, 14 SNPs showed an effect size of 40% or more for rooting percentages after 6 weeks. Among these, SNP RhK5_11050_203Q (P05, effect 41.4%) is associated with the coding region for *GTE4*, a gene involved in cell division activities in meristems, and whose knockout mutants showed reduced lateral root formation [79]. Interestingly, nine of the 14 SNPs with the highest effect sizes were associated with the same contig, namely, RhK5_69. This contig was derived from an EST for a homologue of a probable phosphoinositide phosphatase *SAC9*.

The *SUPPRESSOR OF ACTIN* (*SAC*) gene family comprises nine different genes in *A. thaliana*, divided into three clades (*SAC1-SAC5*, *SAC6-SAC8*, *SAC9*) [80]. Recently, these phosphatases and their substrates – the cell membrane-located phosphoinositides (also known as PIs or PtdInsPs) – were mentioned to be involved in the regulation and signalling of different developmental and growth processes in plants by addressing cellular processes such as endocytosis, vacuolar trafficking, and actin dynamics [80]. Lack of function of different *SAC* phosphatases leads to accumulation of PIs and altered phenotypes [81, 82]. *Sac6-sac8* loss-of-function mutants in *A. thaliana* showed defects in embryo and seed development associated with delayed trafficking of the *PIN1* auxin efflux carrier and decelerated auxin distribution [82]. This led to the assumption that genes of the *AtSAC6-AtSAC8* clade could be involved in auxin-controlled processes [82]. Furthermore, *sac9* mutants showed less formation of lateral roots and cell wall alterations in the form of wall ingrowths into cells of *A. thaliana* seedling primary roots [81, 83].

Recently, Lebecq et al. [84] showed that *sac9* loss-of-function resulted in reduced clathrin-mediated endocytosis, through which *PIN* protein translocation also occurs. In rose, Nguyen et al. [7] identified a SNP (RhK5_827_547Q) on Chr07 at 49.3 Mbp in a gene encoding a phosphoinositide phosphatase family protein that was significantly associated with total root length *in vivo*. Overall, these observations suggest that members of the *SAC* gene family play roles in the regulation of AR formation.

## Marker–trait associations for root number and root fresh mass

Distinct genomic regions (peaks) were identified to be associated with both traits of root number and root fresh mass. Filtering based on the allele configuration effects was applied to three peaks for root number (P04, P06nr, P07) and four peaks for root fresh mass (P03fm, P04, P06fm, P07), with P04 and P07 as overlaps (Fig 5, Tables 2 and 3). The positions of three peaks (peaks P03fm, P06fm, and P07) exhibit agreement with peaks identified for *in vitro* root length, *in vivo* root number, and *in vivo* root dry mass in the study by Nguyen et al. [7]. Several genes known to be involved in AR formation and reviewed by Guan et al. [56], Druege et al. [10], or Li [57] colocalised with the detected peaks in the present study (S8 Table). Among them are the ethylene receptor *ERF1* (P03fm, P07); auxin-associated genes such as *YUC* (P06nr, P06fm), *IAA14* (P06nr), and *LAX3* (P06fm); carbohydrate-associated genes such as *SUC* (P06fm), *HXK3* (P06fm), and *TPP* (P06fm, P07); and genes involved in cell division processes such as *BBM* (P03fm), *CYCs* (P06nr, P06fm), *RID2* (P06nr, P07), and especially *LBD16* and *LBD29*, which were both recently assigned to a conserved superlocus inducing shoot-borne root initiation in tomato [85].

Furthermore, for root number and root fresh mass, SNPs with a high allele configuration effect size were also selected. For root number as well as for root fresh mass, SNP RhK5_19325_1901P (P07) expressed/displayed the highest effect size and was assigned to a gene annotated as a *plastidial pyruvate kinase 3*. Pyruvate kinase activity is increased during AR formation [86, 87]. The SNP RhK5_3575_1325Q (P07) was located in the coding region for *GRAS* family transcription factor *SCR6*, and *SCRs* were reported to be involved in AR formation [71]. RhMCRND_23321_187P had an effect size on root fresh mass of 0.22 g and is localised in the coding region of a gene homologue of the *WD repeat-containing protein 44*. WD40 domain-containing proteins have been described to be involved in root growth [88, 89]. Furthermore, a WD40 repeat gene was shown to be downregulated by *miRNA156* [90]. This miRNA impacts AR formation by regulating *squamosa promoter binding protein-like* (*SPL*), a regulator of the juvenile to adult transition [91].

In summary, we identified genetic components for AR formation-related phenotypic traits. Identified distinct peaks and SNPs usually did not reach the significance threshold, and

compared, for example, to the results of marker–trait association for fragrance in rose [36], we observed more but less prominent peaks. This could indicate that AR is regulated by a higher number of loci, each with smaller effects. To possibly discover even more distinct peaks, a larger number of genotypes should be used in future studies. However, experiments on AR formation are time-intensive, and GWAS approaches are still costly. Lower prices for sequencing and new automated phenotyping approaches offer the possibility to reach this goal in the future. Nevertheless, SNPs in several known AR formation-related genes were identified in our study, but SNPs in genes of unknown function were also found, which may allow further insights into the process of AR formation in the future when gene identity will have been resolved. Interestingly, this study suggests a new link between phosphoinositides and AR formation, and the SNP RhK5_69_1627P within the probable gene *SAC9* was selected to develop a KASP marker.

### Validation of SNP RhK5_69_1627 at the *SAC9* locus

For later marker-assisted selection, a KASP marker was developed based on the SNP RhK5_69_1627P associated with a coding region for a putative *SAC9* phosphoinositide phosphatase. The SNP displayed high and significant additive dosage effects for AR formation percentages, with a high effect size of a 44.7% difference between homozygous ADGs after 6 weeks of cultivation (Fig 6, Table 1). Additionally, a comparatively high effect of 4.5 more roots for quadruplex genotypes compared to nulliplex genotypes was observed. The KASP marker was successfully applied for the two population sets of cut and garden roses (S3 Fig). Minor differences between KASP and chip genotyping results could be due to technical reasons (ST1B in S1 Table). The KASP marker was used in an independent set of 377 genotypes to select homozygous genotypes. A significant effect of allele dosage configuration between 15 nulliplex and 28 quadruplex genotypes was confirmed and reached 32.6% after 6 weeks (Fig 7). Overall, rooting percentages were lower for the KASP marker validation experiments compared to the phenotyping experiments for GWAS. In addition to the genotypic capability to form ARs, these differences could also exist because the cuttings for the KASP verification were harvested from field-grown plants, whereas the stock plants for the GWAS experiments were grown under greenhouse conditions [6].

### Conclusion

Our GWAS on AR formation in cut and garden roses revealed several genomic regions with a significant influence on AR-associated traits. The variability of phenotypic data indicates a larger effect of the environment on AR compared to other rose characteristics, such as petal numbers. Although almost none of the individual markers reached the adjusted significance threshold levels, we could nevertheless identify and validate several markers with large effects on the AR formation-related traits from a number of prominent peaks. All of these peaks colocalise with genes reported to be related to root formation in other plant species. Some of our new markers display large effects on the phenotypes and might be used for marker-assisted selection of parental clones with higher rooting potential in breeding programs where rooting capacity is an important breeding goal.

Furthermore, we developed and verified a KASP marker located in a putative phosphoinositide phosphatase. Taking into account the involvement of *SAC* genes in developmental processes, particularly *SAC6-8* in auxin homeostasis through *PIN* regulation and *SAC9* in endocytosis, further functional analyses to investigate the possible role of *SAC9* in AR formation either in rose or in the model plant *A. thaliana* should follow.

## Supporting information

**S1 Fig. Hydroponic system setup for phenotyping AR formation in rose cuttings over 6 weeks of cultivation.** (A) Set-up of plastic plates with holes to hold rose cuttings placed on black trays filled with tap water (for two weeks) or fertiliser solution. (B) Top and bottom views of cuttings after 6 weeks of cultivation. Scale bars indicate 10 cm.
(TIF)

**S2 Fig. Effects of allele dosage configuration for SNP RhK5_4872_1159Q on AR formation percentages after 3 and 6 weeks.** Allele dosage effects for the SNP RhK5_4872_1159Q on AR formation percentages after 3 (A) and 6 weeks (B) for 95 cut and 95 garden roses. X-axis values show the dosages for the SNP allele from nulliplex (0) to quadruplex (4), where the number of individuals per ADG is given by *n*. Letters indicate significance groups with respect to Fisher´s LSD criterion for $p < 0.05$ under consideration of the Holm–Bonferroni adjustment.
(TIF)

**S3 Fig.** Genotyping results in the cut rose (A) and garden rose sets (B) with the KASP marker assigned to SNP RhK5_69_1627P. The X-axis indicates HEX fluorescence intensity (Allele 1), and the Y-axis indicates FAM fluorescence intensity (Allele 2). The white and darkest red dots represent the homozygous individuals, and blended colours represent the heterozygous individuals. Black crosses indicate the water control, and genotypes with undetermined allele dosages are displayed in grey. The ADGs were determined from the fluorescence signal ratios by using fitTetra [43].
(TIF)

**S4 Fig. Genotyping results with the KASP marker assigned to SNP RhK5_69_1627P applied to an independent set of 377 genotypes.** The X-axis indicates HEX fluorescence intensity (Allele 1), and the Y-axis indicates FAM fluorescence intensity (Allele 2). The white and darkest red dots represent the homozygous individuals, and the blended colours represent the heterozygous individuals. Black crosses indicate the water control, and genotypes with undetermined allele dosages are displayed in grey. The ADGs were determined from the fluorescence signal ratios by using fitTetra [43].
(TIF)

**S1 Table. List of cut and garden rose genotypes tested within AR formation experiments for marker–trait association studies, number of tested cuttings per genotype, and allele configuration for SNP RhK5_69_1627P genotyped with Axiom 68 K array and assigned KASP assay.** (A) 95 cut rose (CR) and 96 garden rose genotypes (GR) tested for marker–trait association for AR formation traits. "*X*" indicates whether the genotype was tested within the distinct experiments (GR1, GR2, CR1, CR2). Allele configurations are displayed as tetraploid numeric values from 0 (nulliplex for allele A) to 4 (quadruplex for allele A). (B) Comparison of homozygous genotypes for SNP RhK5_69_1627P genotyped with the 68 K WagRhSNP Axion chip assay or the KASP assay. Differences in allele configuration are displayed in brackets and in italics.
(XLSX)

**S2 Table. List of 377 independent genotypes available at the Europa-Rosarium Sangerhausen and allele configuration for SNP RhK5_69_1627P genotyped in the KASP assay.**
(XLSX)

**S3 Table. Temperatures measured inside the foil tent during rooting experiments for GWAS analyses (CR1, CR2, GR1, GR2) and KASP marker validation (KASP1, KASP2).**
(XLSX)

**S4 Table. The results of statistical analyses for ANOVA/deviance analyses for factors experimental repetitions, genotype, and panel, as well as mean values, number of cuttings and number of genotypes per panel (A), pairwise comparisons of experiments (B), pairwise comparisons of panels (C), and Pearson´s correlation coefficients (D).**
(XLSX)

**S5 Table. Phenotypic data for AR formation traits collected for 95 cut and 95 garden rose genotypes.** Weighted means and standard deviations for AR formation percentages after 3 to 6 weeks in % (A-D) and number of surviving cuttings, number of rooted cuttings, and number of blocks with surviving cuttings per week. Means and standard deviations for root number and root fresh mass per rooted cutting per genotype for genotypes with n≥3 rooted cuttings (D).
(XLSX)

**S6 Table. Calculated and model-based effect sizes of selected SNPs located within distinct peaks for AR formation percentages after 6 weeks (A), root number (B) and root fresh mass (C) per rooted cutting.** SNP markers are ordered by their effect size, specified as an additive effect unless otherwise stated. Corresponding gene predictions are based on BLASTx searches on GDR and NCBI. *NA* indicates that no gene of known function could be assigned.
(XLSX)

**S7 Table. Allele configuration of homozygous genotypes for SNPs located on contig RhK5_69.**
(XLSX)

**S8 Table. Predicted gene homologues previously described to be involved in the process of AR formation and located within defined regions of distinct peaks for AR formation percentages after 6 weeks, root number and root fresh mass per rooted cutting.** AR formation-associated genes were collected from reviews by Druege et al. [10], Guan et al. [56], and Li [57]. Gene sequences were mapped against the 'Old Blush' rose genome sequence [30].
(XLSX)

**S9 Table. Flanking sequences of SNP RhK5_69_1627P and sequences for derived KASP assay primers.**
(XLSX)

**S10 Table. Phenotypic data for AR formation traits for selected genotypes homozygous for the KASP marker derived from SNP RhK5_69_1627P.** Phenotypic data for 43 homozygous genotypes for SNP RhK5_69_1627P selected with the derived KASP assay in an independent population at Europa-Rosarium Sangerhausen. Weighted means and standard deviations for AR formation percentages after 3 to 6 weeks in % (A-D) and number of surviving cuttings, number of rooted cuttings, and number of blocks with surviving cuttings per week. Genotypes highlighted in red were not taken into account due to too few or no surviving cuttings after 6 weeks.
(XLSX)

## Acknowledgments

The authors would like to thank Rosen Tantau, in particular Jens Krüger, employees from the Federal Plant Variety Office, in particular Dr. Daniela Christ, and employees from Europa Rosarium Sangerhausen, in particular Thomas Hawel and Gerhild Schulz, for kind support and providing plant material. Additionally, we thank Simon Richartz, gardeners and technical

staff of the Department of Woody Plant and Propagation Physiology, Klaus Dreier and gardeners of the Department of Molecular Plant Breeding, and the team of the Central Experimental Unit of the Faculty of Natural Sciences for technical support. Additionally, we thank Prof. Dr. Thomas Scheper, Dr. Frank Stahl, and Nina Scheler from the Institute of Technical Chemistry for providing and introducing into the PhoenixPure System.

## Author Contributions

**Conceptualization:** David Wamhoff, Thomas Debener, Traud Winkelmann.

**Data curation:** David Wamhoff, Dietmar Frank Schulz, Thomas Debener.

**Formal analysis:** David Wamhoff, Laurine Patzer.

**Funding acquisition:** Dietmar Frank Schulz, Thomas Debener, Traud Winkelmann.

**Investigation:** David Wamhoff, Laurine Patzer.

**Methodology:** David Wamhoff, Laurine Patzer, Dietmar Frank Schulz, Thomas Debener, Traud Winkelmann.

**Project administration:** David Wamhoff, Thomas Debener, Traud Winkelmann.

**Resources:** David Wamhoff, Dietmar Frank Schulz, Thomas Debener, Traud Winkelmann.

**Software:** David Wamhoff.

**Supervision:** Thomas Debener, Traud Winkelmann.

**Validation:** David Wamhoff, Laurine Patzer, Thomas Debener, Traud Winkelmann.

**Visualization:** David Wamhoff, Laurine Patzer, Thomas Debener, Traud Winkelmann.

**Writing – original draft:** David Wamhoff, Laurine Patzer, Thomas Debener, Traud Winkelmann.

**Writing – review & editing:** David Wamhoff, Laurine Patzer, Dietmar Frank Schulz, Thomas Debener, Traud Winkelmann.

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
