## [Decision Letter · Decision Letter 0]

18 May 2023

PONE-D-23-09430GWAS of adventitious root formation in roses identifies a putative phosphoinositide phosphatase (SAC9) for marker-assisted selectionPLOS ONE

Dear Dr. Wamhoff, Thank you for submitting your manuscript to PLOS ONE. After careful consideration, we feel that it has merit but does not fully meet PLOS ONE’s publication criteria as it currently stands. Therefore, we invite you to submit a revised version of the manuscript that addresses the points raised during the review process.

We look forward to receiving your revised manuscript.

Kind regards,

Md Ashrafuzzaman, Ph.D.

Academic Editor

PLOS ONE

Journal Requirements:

"..This study was carried out with financial support from Leibniz University Hannover."

"TD: Aif Projekt GmbH (ZF4619701MD8), https://www.aif-projekt-gmbh.de/

TD: Federal Institute of Agriculture and Nutrition (BLE) (FKZ: 2819HS010), https://www.ble.de/

6. Please include a new copy of Table 1 in your manuscript; the current table is difficult to read. Please follow the link for more information: https://blogs.plos.org/plos/2019/06/looking-good-tips-for-creating-your-plos-figures-graphics/

**Additional Editor Comments:**

I have completed my evaluation of your manuscript. The reviewers recommend reconsideration of your manuscript following minor revision. I invite you to resubmit your manuscript after addressing the comments below.

Reviewers' comments:

Reviewer's Responses to Questions

**Comments to the Author**

1. Is the manuscript technically sound, and do the data support the conclusions?

Reviewer #1: Yes

Reviewer #2: Yes

2. Has the statistical analysis been performed appropriately and rigorously? 

Reviewer #1: Yes

Reviewer #2: Yes

3. Have the authors made all data underlying the findings in their manuscript fully available?

Reviewer #1: Yes

Reviewer #2: Yes

4. Is the manuscript presented in an intelligible fashion and written in standard English?

Reviewer #1: Yes

Reviewer #2: Yes

5. Review Comments to the Author

Reviewer #1: Comments on manuscript PONE-D-23-09430

The manuscript at hand explores GWAS as tool to identify adventitious rooting-related markers for marker-assisted selection in rose. This is an important piece of research that adds to the growing body of evidence towards better propagation strategies and bridging the knowledge gap in adventitious root physiology. I only have a few comments and suggestions that will hopefully be useful in improving the manuscript.

L183: How were these criteria chosen?

L240: Apart from this cultivar, was survival high? By which I mean: did all cuttings that did not root die? Did any undergo callogenesis?

L243: Did any consistent pattern arise when comparing cut and garden roses? This goes for %rooting, root number, and total root mass.

L261: I wonder if a metric of root mass per root could reveal any pattern regarding the trade-off between number and biomass. Number and mass have a strong correlation (.65). I wonder if this is uniformly true across the spectrum of rooting ability. Maybe splitting the rooting data into quartiles and checking correlation coefficients within those could give additional information of the natural variability of this phenotype.

L286-L314: This paragraph is too long and should be broken into smaller ones.

Fig. 4: What was the rationale behind the choice of peak below the threshold for further analysis? For Chrs 02, 03, and 05 one could argue there are other possible choices.

Fig. 5: Same question, specifically regarding Chr05.

Table 3: Use decimal points, not commas, as decimal separators.

L504: Could this not mask some early-stage (3 weeks in, for example) differences that could themselves be correlated to rooting success? Some of the most important events in the regulation of adventitious root formation indeed occur before any roots are visible.

L534: I suggest breaking down this long paragraph for the sake of clarity.

Figure S1: Could the authors add a scale bar to this figure? Specifically for the bottom right panel, so readers can have an idea of root length.

Reviewer #2: The experiments were well designed and carried out. The results were presented and adequately discussed. The manuscript represents a good contribution for both understanding AR formation and breeding for cutting propagation.

6. PLOS authors have the option to publish the peer review history of their article (what does this mean?). If published, this will include your full peer review and any attached files.

Reviewer #1: **Yes: **Johnatan Vilasboa

Reviewer #2: **Yes: **Bisognin, Dilson Antonio

---

## [Author Response · Author response to Decision Letter 0]

1 Jun 2023

Submission ID PONE-D-23-09430

GWAS of adventitious root formation in roses identifies a putative phosphoinositide phosphatase (SAC9) for marker-assisted selection

PLOS ONE

Dear editior Dr. Ashrafuzzaman, dear reviewers Dr. Vilasboa and Prof. Dr Bisognin,

we are very grateful for your valuable time spent into our manuscript and your comments which we carefully considered and followed wherever possible as explained below. Please find our responses indicated in blue below each point raised by one of you. We hope that you will find all points addressed in the right way and look forward to hearing from you again soon.

Thank you and kind regards

David Wamhoff, Laurine Patzer, Dietmar Schulz, Thomas Debener, and Traud Winkelmann

Revision Deadline: Jul 02 2023 11:59PM

Additional Editor Comments:

I have completed my evaluation of your manuscript. The reviewers recommend reconsideration of your manuscript following minor revision. I invite you to resubmit your manuscript after addressing the comments below.

Thanks a lot for the invitation and possibility to do revisions on our manuscript.

Reviewers' comments:

Reviewer's Responses to Questions

Comments to the Author

1. Is the manuscript technically sound, and do the data support the conclusions?

Reviewer #1: Yes

Reviewer #2: Yes

2. Has the statistical analysis been performed appropriately and rigorously?

Reviewer #1: Yes

Reviewer #2: Yes

3. Have the authors made all data underlying the findings in their manuscript fully available?

Reviewer #1: Yes

Reviewer #2: Yes

4. Is the manuscript presented in an intelligible fashion and written in standard English?

Reviewer #1: Yes

Reviewer #2: Yes

5. Review Comments to the Author

Reviewer #1: Comments on manuscript PONE-D-23-09430

The manuscript at hand explores GWAS as tool to identify adventitious rooting-related markers for marker-assisted selection in rose. This is an important piece of research that adds to the growing body of evidence towards better propagation strategies and bridging the knowledge gap in adventitious root physiology.

Thanks a lot for this positive feedback. We are glad that you consider our work as an important contribution for the scientific community regarding physiological and genetic control of adventitious root formation.

I only have a few comments and suggestions that will hopefully be useful in improving the manuscript.

Thank you for your valuable suggestions and comments which indeed helped to improve the manuscript. Changes in the manuscripts are indicated in the following responses your comments. The lines where the changes can be found in the manuscript are based on the file with track changes. Changed passages are indicated in italic within the following answers. 

L183: How were these criteria chosen?

Thank you for the comment and sorry that this was not clear in the submitted manuscript. The criteria were set based on the number of SNPs resulting from setting the effect size threshold. The threshold should not be too high not to overlook SNPs with a high impact, but it should not be too low, to allow focusing on SNPs showing a relative strong effect. However, to address your point, we added a passage to clarify the setting of the criteria in lines 191 ff.: 

• To achieve the main objective of developing a selection marker for AR formation in rose, we defined threshold values for the effect sized for the different phenotypic data. These thresholds were set in a manner that allowed focusing on the SNPs with the most substantial effects within the identified peaks. Thereby, SNPs were selected for further investigation, aligning with the main aim of the study.

L240: Apart from this cultivar, was survival high? By which I mean: did all cuttings that did not root die? Did any undergo callogenesis?

ᶦFrühlingsduftᶦ (FD) was excluded from the analyses due to a low number (n<3) of blocks with surviving cuttings.

This is an important point, thank you. To better describe the fate of the cuttings, we added the number of dead cuttings as well as the surviving percentage for each genotype and week of evaluation in S5_table. Not all cuttings that did not form roots died within the six weeks rooting period. We did not focus on surviving percentage due to numerous reasons: Cutting quality depends on several factors such as season, stock plant constitution, or position of the cutting at the stock plant. To encounter for this variability, we included a sufficiently high number of cuttings per cultivar. Callus formation was observed, but in very different intensities. A correlation of callogenesis and rooting was not observed. Therefore, we decided to not include callogenesis data in this study. 

L243: Did any consistent pattern arise when comparing cut and garden roses? This goes for %rooting, root number, and total root mass.

Thank you for raising this legitimate question. Colour patterns in Fig 1 (rooting%) and Fig 2 (root number, total root mass) indicated, that there was only a minor pattern in response of the garden and cut rose panel. However, we performed additional statistical analyses for rooting% after 3 to 6 weeks, root number per rooted cutting, and root fresh mass. The results were interesting and added to table (S4_Table: ST4A): Because the factor “panel” had a significant effect on rooting after three, four, and five weeks, we also added results of pairwise comparisons in the new S4_Table sheet ST4C, showing that the cut rose panel genotypes rooted in significantly higher frequencies. The previously named sheet ST4C was changed to ST4D, this was changed in the text as well (Lines 282 and 297). 

Furthermore, we adjusted the following text passages (passages given in italics were changed or added): 

• Line 223 ff.: “To test for differences between experiments, (generalised) linear mixed models with the experiment as a fixed effect and genotype and block as random effects were used, including only the 13 garden rose genotypes, which were also tested as references in CR1 and CR2.”

The part in italics was moved here from l.226 f., because we recognised at revising the manuscript that this statement was wrongly placed. We are thankful that your comment enabled us to correct this mistake.

Also corrections were done in l. 225 ff.: “For differences between genotypes and panels, data were analysed separately in (generalised) linear mixed models with genotype or panel as a fixed effect and experiment and block as random effects.”

• Lines 256 f.: “Furthermore, genotypes of the panel CR showed higher rooting than those of the panel GR after 3, 4, and 5 weeks, respectively (ST4C in S4 Table).” 

• Lines 488 ff.: “In addition, significantly higher rooting percentages in panel CR compared to panel GR after 3, 4, and 5 weeks, but not after 6 weeks and not for root number and root fresh mass per rooted cutting led to the conclusion, that the genotypes of the two panels differed in rooting speed, but not in rooting quality. However, these differences could also be due to seasonal differences, as only some of the garden rose genotypes were tested synchronously in same experiments with the cut rose genotypes.”

• Lines 971 f.: “S4 Table. The results of statistical analyses for ANOVA/deviance analyses for factors experimental repetitions, genotype, and panel, as well as mean values, number of cuttings and number of genotypes per panel (A), pairwise comparisons of experiments (B), pairwise comparisons of panels (C), and Pearson´s correlation coefficients (D).”

L261: I wonder if a metric of root mass per root could reveal any pattern regarding the trade-off between number and biomass. Number and mass have a strong correlation (.65). I wonder if this is uniformly true across the spectrum of rooting ability. Maybe splitting the rooting data into quartiles and checking correlation coefficients within those could give additional information of the natural variability of this phenotype.

Thank you for raising this valuable suggestion which we followed: We conducted correlations for the metric values of FM per root. Although the correlations were not particularly strong, we observed a decreasing correlation with rooting percentages after 3, 4, and 5 weeks. Interestingly, we found no correlation with rooting percentages after 6 weeks or root number per rooted cutting. However, there was a significant and strong correlation between total root FM per rooted cutting and FM per root. Furthermore, we noticed a slight but consistent pattern indicating that the correlation between rooting percentage and FM per root becomes stronger and more significant as the cuttings root earlier.

To further investigate your second point, i.e., to analyse correlation coefficients within quartiles, we divided the data based on the rooting data after 6 weeks for a total of 190 genotypes into the four quartiles (Q) as follows: 47 (Q1), 48 (Q2), 48 (Q3), and 47 genotypes (Q4). However, it is important to note that for Q1 only seven genotypes could be used for correlation analyses, as these were the only ones that showed root formation on three or more cuttings. Consequently, the number of data points available for Q1 is very limited when considering parameters such as root number, total root FM, and FM per root. You may please find the results of the analyses in the Figure below. 

We observed relatively high correlations between rooting percentages after 5 and 6 weeks for all quartiles, which further supports our conclusion written in the manuscript that evaluation could be concluded after 5 weeks of rooting. Additionally, we found that correlations between total root FM and root number were significant for Q2, Q3, and Q4, except for Q1. Notably, the correlations between rooting percentages and total root FM, as well as FM per single root, were particularly strong for Q2 and Q3. However, an interesting finding emerged in Q4, where we observed a significant negative correlation between FM per single root and root number. This implies that a lower number of roots per rooted cutting corresponds to a higher FM per single root or vice versa.

In summary, dividing the data into quartiles delivered few further insights. However, we think that this additional information may take attention from the main objectives of this manuscript which is already rather long. Therefore, we have decided to share the detailed results with you in this response letter (provided below). Nevertheless, to address your suggestion regarding the FM per root we made adjustments to Fig 3 by including correlation coefficients for FM per root. Furthermore, we have also revised the following text passages to ensure clarity and accuracy:

• Lines 147 f.: Pearson´s correlation coefficients were calculated between AR formation percentages after 3, 4, 5, and 6 weeks of rooting, as well as for root number and total root fresh mass per rooted cutting, and the calculated average root fresh mass per root.

• Lines 282 & 286: “root fresh mass” was changes to “total root fresh mass”

• Lines 287 ff.: The calculated average fresh mass per root was correlated significantly with rooting percentages after 3 (0.31), 4 (0.3), and 5 weeks (0.21) in a decreasing manner, but not with rooting percentages after 6 weeks (Fig 3). The correlation with total root fm was high (0.63) (Fig 3).

• Lines 292 ff.: Pearson´s correlation coefficients between AR formation percentages after 3, 4, 5, and 6 weeks, root number (root nr), total root fresh mass (total root fm), and calculated average fresh mass per root (fm per root) per rooted cutting for 190 genotypes (correlations between AR formation %) or 144 genotypes (correlations involving root number and root fresh mass).

• Lines 495 f.: However, the calculated average fresh mass per root was not significantly correlated with the parameters rooting percentages after 6 weeks and root number.

• Lines 503 ff.: The correlations between FM per root and rooting percentages indicate that there may be a link between growth time after root penetration and the qualitative constitution of a single root.

L286-L314: This paragraph is too long and should be broken into smaller ones.

You are completely right that two shorter paragraphs would improve readability. We started a new paragraph in line 314.

Fig. 4: What was the rationale behind the choice of peak below the threshold for further analysis? For Chrs 02, 03, and 05 one could argue there are other possible choices.

This is indeed a reasonable question which we will try to explain here, and in the revised manuscript: Choosing the three mentioned peaks for rooting% after 6 weeks in Fig 4 was based on the consideration of several factors: First, the peak regions were chosen regarding their proximity to the threshold. In addition, we took into account the density of the peaks (no “loose” SNP clusters) and the presence of the peaks for other time points or phenotypic parameters.

To clarify this point in the manuscript, we added the following statements: 

• Lines 167 ff.: Peaks were selected based on the following criteria: low distance to the significance threshold, low distance of the single SNPs within the SNP cluster to each other, and appearance of the peak for several AR formation traits. Identified peaks were analysed in more detail to detect and select SNPs with large effects on AR formation traits […].

• Lines 316 ff.: These are indicated in Fig 4 as peaks P02 (Chr02, 0-10 Mbp, also present after 4 and 5 weeks), P03 (Chr03, 30-43 Mbp, also present after 5 weeks) and P05 (Chr05, 18-28 Mbp, also present after 3 and 5 weeks), and were the peaks to the p-value threshold.

Fig. 5: Same question, specifically regarding Chr05.

The argument for deciding on the designated peaks in Fig 5 were the same as explained above for Fig 4: P03fm is a dense peak that clearly stands out from the background noise, P04 shows relatively low p-values for both root-nr and root-fm and interestingly also appears for rooting after 4 weeks, P06nr is a very prominent peak relatively close to the threshold p-value, just like P06fm which also appears as a prominent peak for rooting% after 5 and 6 weeks, P07 is a very prominent peak for both root-nr and root-FM. 

Table 3: Use decimal points, not commas, as decimal separators.

Thank you for the thorough checking and sorry for inaccuracy and inconsistence. We changed commata to points.

L504: Could this not mask some early-stage (3 weeks in, for example) differences that could themselves be correlated to rooting success? Some of the most important events in the regulation of adventitious root formation indeed occur before any roots are visible.

You are right, it is fully true, that some of the most important processes in adventitious root formation happen and are regulated even long before the outgrowth of the roots. However, to address differences in the earlier processes during induction and initiation, different methods like histology at early time points or RNA seq - to mention only two possibilities - would be needed and should definitely follow. However, correlations in Fig 3 combined with raising total number of rooted cuttings from week3 to week6 (see S5_Table) show, that later time points are referring to final rooting success in a better way than those of earlier time points. Thus, to our opinion they better reflect the rooting response/ability of the genotype, for which we intend to develop markers. 

L534: I suggest breaking down this long paragraph for the sake of clarity.

Yes, you are completely right. We added two new paragraphs in line 564 and line 576. 

Figure S1: Could the authors add a scale bar to this figure? Specifically, for the bottom right panel, so readers can have an idea of root length.

Thanks for this very good suggestion. We added a scale bar to the two images in b. We think that a scale in the two upper images would be misleading due to the angle of view and spatial dimensions in those images. 

We uploaded S1_Fig as a corrected version and adjusted the caption of the supplemental figure in line 938.

Reviewer #2: The experiments were well designed and carried out. The results were presented and adequately discussed. The manuscript represents a good contribution for both understanding AR formation and breeding for cutting propagation.

Thank you very much for taking time to review our submitted manuscript as well as for your positive opinion on our manuscript. We are very satisfied that you see a good contribution for the addressed scientific topics in our manuscript.

Additional information:

We updated Fig1 and increased it in quality.

---

## [Decision Letter · Decision Letter 1]

6 Jun 2023

GWAS of adventitious root formation in roses identifies a putative phosphoinositide phosphatase (SAC9) for marker-assisted selection

PONE-D-23-09430R1

Dear Dr. David Wamhoff,

I am pleased to inform you that your manuscript has been judged scientifically suitable for publication and will be formally accepted for publication once it meets all outstanding technical requirements.

Kind regards,

Md Ashrafuzzaman, Ph.D.

Academic Editor

PLOS ONE

Additional Editor Comments (optional):

Reviewers' comments:

Reviewer's Responses to Questions

**Comments to the Author**

1. If the authors have adequately addressed your comments raised in a previous round of review and you feel that this manuscript is now acceptable for publication, you may indicate that here to bypass the “Comments to the Author” section, enter your conflict of interest statement in the “Confidential to Editor” section, and submit your "Accept" recommendation.

Reviewer #1: All comments have been addressed

2. Is the manuscript technically sound, and do the data support the conclusions?

Reviewer #1: Yes

3. Has the statistical analysis been performed appropriately and rigorously? 

Reviewer #1: Yes

4. Have the authors made all data underlying the findings in their manuscript fully available?

Reviewer #1: Yes

5. Is the manuscript presented in an intelligible fashion and written in standard English?

Reviewer #1: Yes

6. Review Comments to the Author

Reviewer #1: All raised questions were addressed with rigour and, whenever relevant, changes were implemented in the revised manuscript.

I congratulate the authors for their diligence and openness to suggestions.

7. PLOS authors have the option to publish the peer review history of their article (what does this mean?). If published, this will include your full peer review and any attached files.

Reviewer #1: **Yes: **Johnatan Vilasboa

---

## [Editor Report · Acceptance letter]

10 Aug 2023

PONE-D-23-09430R1 

GWAS of adventitious root formation in roses identifies a putative phosphoinositide phosphatase (SAC9) for marker-assisted selection 

Dear Dr. Wamhoff:

I'm pleased to inform you that your manuscript has been deemed suitable for publication in PLOS ONE. Congratulations! Your manuscript is now with our production department. 

Kind regards, 

on behalf of

Dr. Md Ashrafuzzaman 

Academic Editor

PLOS ONE